# Self healable neuromorphic memtransistor elements for decentralized sensory signal processing in robotics

Rohit Abraham John [1], Naveen Tiwari[1], Muhammad Iszaki Bin Patdillah[2], Mohit Rameshchandra Kulkarni[1], Nidhi Tiwari[2], Joydeep Basu [3], Sumon Kumar Bose [3], Ankit [1], Chan Jun Yu[4], Amoolya Nirmal[1], Sujaya Kumar Vishwanath[1], Chiara Bartolozzi [5], Arindam Basu [3]✉ & Nripan Mathews [1,2]✉

Sensory information processing in robot skins currently rely on a centralized approach where signal transduction (on the body) is separated from centralized computation and decision-making, requiring the transfer of large amounts of data from periphery to central processors, at the cost of wiring, latency, fault tolerance and robustness. We envision a decentralized approach where intelligence is embedded in the sensing nodes, using a unique neuromorphic methodology to extract relevant information in robotic skins. Here we specifically address pain perception and the association of nociception with tactile perception to trigger the escape reflex in a sensorized robotic arm. The proposed system comprises self-healable materials and memtransistors as enabling technologies for the implementation of neuromorphic nociceptors, spiking local associative learning and communication. Configuring memtransistors as gated-threshold and -memristive switches, the demonstrated system features in-memory edge computing with minimal hardware circuitry and wiring, and enhanced fault tolerance and robustness.

[1] School of Materials Science and Engineering, Nanyang Technological University, 50 Nanyang Avenue, Singapore 639798, Singapore. [2] Energy Research Institute @ NTU (ERI@N), Nanyang Technological University, Singapore 637553, Singapore. [3] School of Electrical and Electronic Engineering, Nanyang Technological University, 50 Nanyang Avenue, Singapore 639798, Singapore. [4] School of Mechanical and Aerospace Engineering, Nanyang Technological University, 50 Nanyang Avenue, Singapore 639798, Singapore. [5] Event-Driven Perception for Robotics, Italian Institute of Technology, via San Quirico 19D, 16163 Genova, Italy. ✉email: arindam.basu@ntu.edu.sg; nripan@ntu.edu.sg

Conventional robots carry out tasks in a structured programmable manner in controlled environments. The next generation of robots will work in unconstrained dynamic environments and in close proximity with humans. In this scenario, robots must be able to correctly perceive the external world and adapt their behaviour accordingly. Tactile and somatosensory perception are crucial to handle physical contact and develop spatial perception, to plan movements that avoid contact with obstacles[1]. However, few robotic platforms are equipped with tactile sensors to match the scale of the human skin (the majority only in their end-effectors and a small percentage over their full body[2]-HEX-O-SKIN[3] and RoboSkin[4]), as covering robots with a myriad of small sensing sites brings about a number of technological challenges ranging from wiring to sensitivity, bandwidth, fault tolerance and robustness[5,6]. The current approach in robotics is based on the use of remote embedded sensors (on the robot body) and central processing. These implementations do not match the scale, connectivity and energy efficiency of their biological counterpart and often place little emphasis on fault tolerance against mechanical damage, which is essential for robots working in uncertain environments.

Implementations relying on the serial transmission of sensory information result in latency bottlenecks directly proportional to the number of sensors[6]. Time-divisional multiple access[7,8] protocols can reduce the wiring complexity but require sequential and periodic sampling of sensors to map the data distribution, hence falling behind on latency and scalability. Inspired from biology, there have been efforts to mimic the signal processing and data handling architectures of the human nervous system, relying on neuromorphic, event-driven sampling of the tactile signal and asynchronous communication (e.g. address event representation[6,9] and spread spectrum techniques[10]). So far, such systems are limited to encoding sensor stimuli with trains of digital pulses for simple signal transduction[11] and high compression of the tactile signals, tackling the issues of wiring, communication bandwidth and efficiency to some extent. However, they still rely on the transfer of low-level sensory information to centralized neuromorphic spike-based processing units[12–14] to use biologically inspired learning methodologies to extract relevant information from the tactile signals, hence becoming vulnerable to the above-mentioned issues. Additionally the robustness of such demonstrations to inadvertent mechanical damage and harsh operating conditions is hitherto unaddressed. Most critically, from the device perspective, current neuromorphic implementations are based on complementary metal-oxide-semiconductor (CMOS) technology that does not match the scale and connectivity of the human nervous system. For example, the implementation of a single CMOS-based synapse with spike-timing-dependent plasticity (STDP)-based learning rule requires multiple transistors, resistors and capacitors, which limits its scalability[15]. Memristive devices, instead, enable power-efficient in-memory computations via neurons and synapses with much simpler circuitry[16–19].

We present a unique, decentralized neuromorphic decision-making concept to lower the temporal redundancy of event-based sensory signals and vastly reduce the amount of data shuttled to the central processing system, hence lowering the latency and wiring demands[6]. We enable this based on the unique sliding threshold behaviour of biological nociceptors and associative learning in synapses to shift intelligence to the location of the sensor nodes. We demonstrate a three-tier decision-making process flow-nociceptors identify and filter noxious information based on short-term temporal correlations, synapses associatively learn patterns in sensory signals with noxious information and neurons integrate synaptic weights. In this work, we propose and demonstrate the first comprehensive memristive implementation of neuromorphic tactile receptors and their fusion with memristive synapses and CMOS neurons. We present three-terminal indium–tungsten oxide (IWO) memtransistors with ionic dielectrics as peripheral signal processors which when fashioned as volatile gated-threshold elements emulate artificial nociceptors and when fashioned as non-volatile gated-memristive switches emulate artificial synapses. Enabled by the computational and circuit efficiency of these memtransistors, we optimize wiring, data transfer and decision-making latency by decentralizing tactile signal processing to the transduction sites. While the proposed method is different from biology where millions of nerve bundles connect the peripheral nervous system to the central nervous system, it is a necessary solution for artificial robotic nervous systems where it is impossible to replicate the wiring density seen in biology. To demonstrate the feasibility of the proposed approach, we develop a proof-of-concept system comprising artificial nociceptors that respond to pain produced by touching a sharp tip and receptors that respond to pressure and temperature, coupled with spiking memristive learning synapses and CMOS neurons to associate pressure and pain perception. In comparison to the artificial afferent nerve implementations utilizing oscillators, Mott memristors[20] and organic synaptic transistors[21] with physically separate signal transduction and processing, we propose a unique, decentralized scheme and demonstrate decision-making at the sensor node as a viable solution to address the peripheral sensory signal processing in robotics.

## Results

Figure 1 illustrates the concept of decentralized intelligence for robotics. In the conventional centralized approach, signal transduction is separated from centralized computation and all the learning happens at a powerful central processor. In comparison, in the proposed decentralized approach, learning is embedded into the sensor nodes, reducing the wiring complexity at the same time improving latency and fault tolerance. Nociceptors identify and filter noxious information based on short-term temporal correlations, synapses associatively learn patterns in sensory signals with noxious information and neurons integrate synaptic weights. Nociception is implemented by satellite threshold adjusting receptors (STARs). Built with memtransistors that act as gated-threshold switches, STARs possess unique features of no adaptation, relaxation and sensitization, and differs from other common sensory receptors in recalibrating their threshold and response only upon injury[22]. Associative learning is implemented in satellite learning modules (SLMs) near the sensing nodes, composed of satellite weight adjusting resistive memories (SWARMs) and CMOS satellite spiking neurons (SSNs). Configuring memtransistors as gated-memristive switches, learning in the SLMs occur via strengthening and weakening of the memristive connections between spiking neurons using STDP[23]. This increases the tolerance to nociceptor damage while reducing the number of wires to be connected to upstream processors. Robustness to inadvertent mechanical damage and harsh operating conditions is further enhanced by the self-healing capabilities of the active switching devices (STARs and SWARMs), going beyond healing at only the substrate level[10]. We report the first healable neuromorphic memristive devices—artificial nociceptors and synapses, to the best of our knowledge. The resulting system is integrated on a robotic hand, the detection of pain signals and the perception of the corresponding pressure by mechanoreceptors triggers an avoidance reflex by the robotic arm. Our results illustrate the viability of implementing the reflex locally with improved latency and fault tolerance rather than in a centralized processing unit.

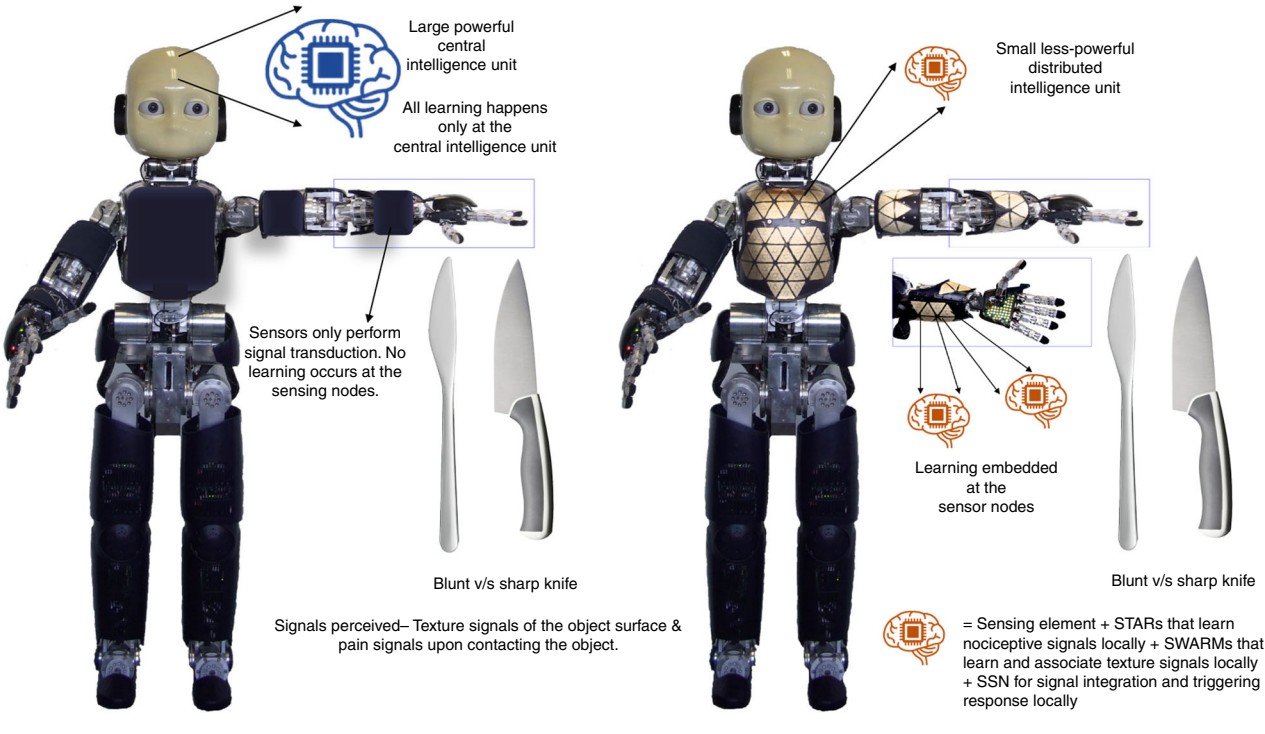

Centralized intelligence

Decentralized intelligence

**Fig. 1 Concept illustration of centralized and decentralized intelligence in robotics.** In the centralized approach, sensing elements are decoupled from the signal processing circuitry. All the learning happens at a powerful large central processor. In comparison, in the proposed decentralized approach, learning is embedded into the sensor nodes, reducing the wiring complexity at the same time improving latency and fault tolerance. In this work, pressure signals from mechanoreceptors are processed by small distributed intelligence units—each comprising a sensing element, satellite threshold adjusting receptors (STARs) that learn nociceptive or pain signals locally, satellite weight adjusting resistive memories (SWARMs) that learn texture signals locally and establish association between the texture and nociceptive signals, and satellite spiking neurons (SSNs) that integrate synaptic weights.

**Satellite threshold adjusting receptors.** With a highly distributed set of receptors, sensory/afferent and motor/efferent nerves, the peripheral nervous system in humans processes lower-order information, signals and relays information between the central nervous system and other areas of the body, which allows us to react to our environment[24]. Correspondingly, a robotic nervous system should offer a rich peripheral interface to accommodate various sensory receptors, locally process lower-order sensory information and relay necessary higher-order signals to the main learning modules to accelerate inference and decision-making.

Modelled on the working of biological nociceptors, STARs act as peripheral lower-order signal processing units that detect noxious stimuli. Located at the end of sensory neuron's axon, biological nociceptors transmit warning action potentials to the central nervous system upon arrival of noxious stimuli such as mechanical stress and temperatures above the pain threshold. Unlike other sensory receptors that adapt their sensitivity upon continuous exposure to stimuli, nociceptors operate uniquely in two modes, exhibiting an intensity-dependent sliding threshold behaviour. In their normal operation mode (Mode-1: defined as the absence of noxious stimuli), the nociceptors maintain a constant threshold and do not adapt to stimuli. Upon injury, they enter an emergency mode (Mode-2) and recalibrate their threshold and response, exhibiting features of relaxation and sensitization to overprotect the injured site. In Mode-2, the sensitivity threshold to stimuli decreases, enhancing the response to innocuous stimuli immediately following noxious stimuli and decreasing latency. Switching between these modes, the sliding threshold function allows to filter significant noxious information from other sensory information, while the short-term memory features of relaxation and sensitization fuses temporal

correlations with noxious information to enable lower-order processing of sensory signals. Figure 2a illustrates the working principle of biological nociceptors and the analogy to STARs. In the system we propose, pressure stimuli from mechanoreceptors or thermal stimuli from thermal receptors represent the signal of interest analogous to noxious inputs in biology. As a first demonstration of this concept, we implement STARs by configuring IWO thin film transistors (Supplementary Note 1, Supplementary Fig. 1) to operate as gated-threshold switches. In the gated-threshold a.k.a. diffusive mode, migration and relaxation of ions in the ionic dielectric temporarily strengthens and weakens the charge carrier accumulation in the semiconducting channel, resulting in a volatile hysteresis, as detailed in Supplementary Note 2. This ion migration-relaxation dynamics at the semiconductor–dielectric interface defines the volatile short-term memory/plasticity behaviour in our memtransistors and is harnessed to present the temporal dynamics of artificial nociceptors or STARs.

To demonstrate the unique sliding threshold feature, the STARs are pulsed with voltage triggers representing external stimuli and the corresponding output current responses are recorded as a function of time. While voltage triggers of weak amplitude ($V_{gs} = 1\,V$) are used to generate normal-state responses (no pain), intense voltage shocks of high amplitude ($V_{gs} \geq 2\,V$) representing pain/injury (noxious inputs) generate sensitized responses from STARs. In their normal state (Mode-1), the STARs generate output responses but is unable to reach the pain threshold ($I_{nox} = 3.3\,mA$) even when stimulated by a train of voltage triggers (number ~50). Increased amplitude, pulse width and number of triggers enhances the current response and reduces the incubation time (time required to reach the pain

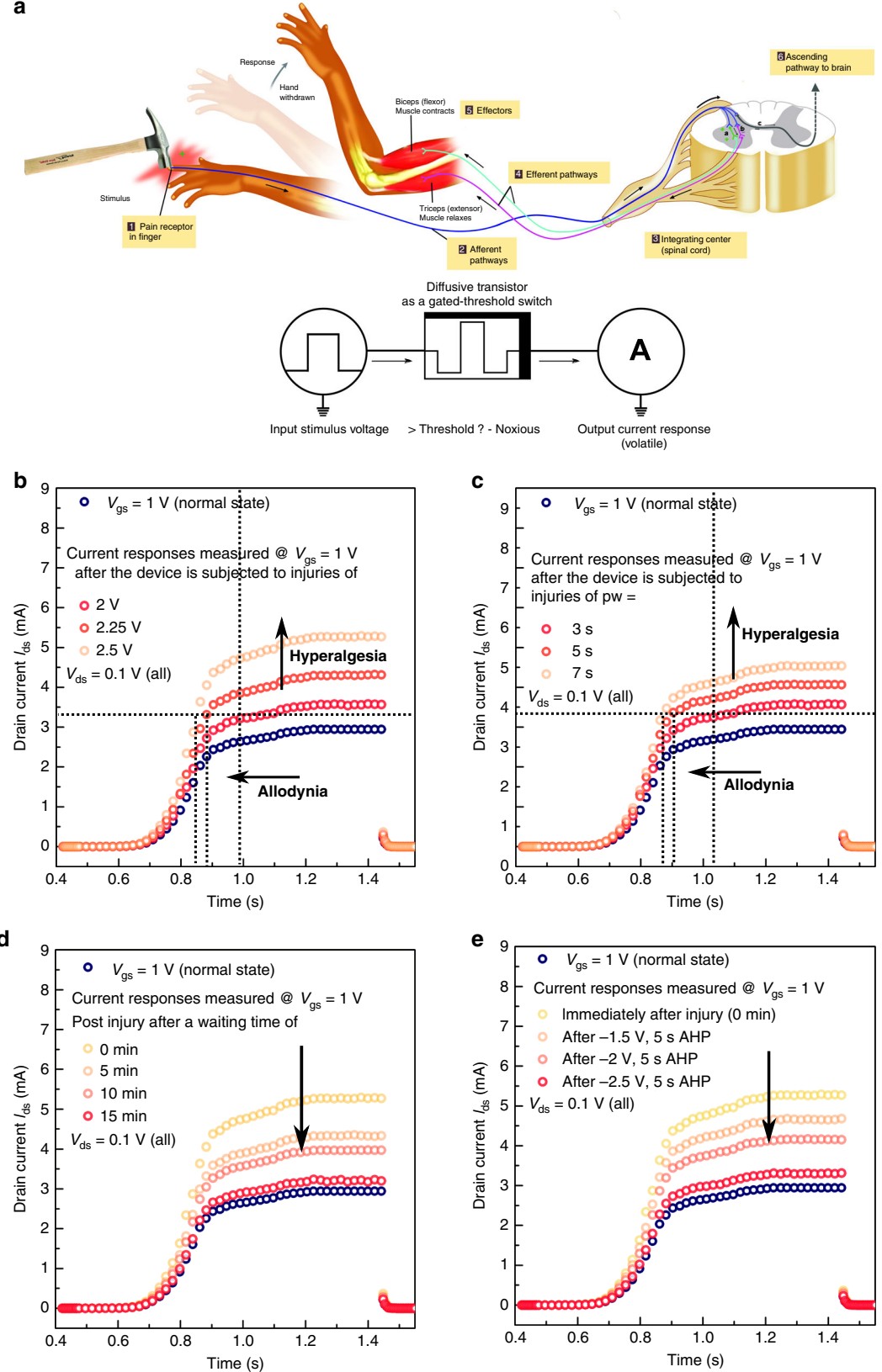

threshold) as shown in Supplementary Note 2 and Supplementary Fig. 2, akin to biological nociceptors. Saturation of current responses upon persistent activation and volatility of these states helps maintain a constant pain threshold throughout the normal-state operation. Supplementary Note 2, Supplementary Figs. 3

and 5 show the relaxation effect of STAR during a stimulation protocol with paired pulses at different interval times (5–32 ms): the priori triggers ($V_1 = 1.2$ V) are of noxious type, generating pain response, while the successive triggers ($V_2 = 0.75$ V) are innocuous stimuli that would not elicit a response. As expected,

**Fig. 2 Nociceptive signal processing in satellite threshold adjusting receptors (STARs). a** Working principle of biological nociceptors and the analogous STARs. Upon arrival of a noxious stimulus with intensity above the pain threshold, the nociceptor generates and relays action potentials to the brain for further processing. Similarly, voltage pulses applied on the STAR generate significant current outputs above the threshold voltage of the transistor. **b**–**e** Voltage triggers of weak amplitude ($V_{gs} = 1$ V) are used to generate normal-state responses, while intense voltage shocks of high amplitude ($V_{gs} \geq$ 2 V) representing injury generate sensitized responses. Normal-state responses are initially measured at a $V_{gs} = 1$ V. Noxious stimuli ($V_{gs} \geq 2$ V) are next applied on the gate terminal of the STARs, after which the sensitized current response curves are once again measured at $V_{gs} = 1$ V. Increased amplitude **b** and pulse width **c** of the noxious stimuli enhance the current response (hyperalgesia) and reduce the incubation time/threshold (allodynia), akin to biological nociceptors. The device is subjected to injuries represented by **b** voltage shocks of 2, 2.25 and 2.5 V, pulse width = 3 s and **c** voltage shocks of 2 V, pulse width = 3, 5 and 7 s. The threshold switching behaviour also enables **d** passive healing with time and **e** active healing with curing pulses of opposite polarity. For **d**, the device is subjected to an injury of 2.5 V, pulse width = 3 s. The responses are measured post-injury at $V_{gs} = 1$ V after waiting for 5, 10 and 15 min respectively. For **e**, the device is subjected to an injury of 2.5 V, pulse width = 3 s. Next, active healing pulses (AHP) of −1.5, −2 and −2.5 V are applied for 5 s and the responses are measured post-healing at $V_{gs} = 1$ V.

innocuous stimuli that arrived more than 10 ms after the $V_1$ stimuli, i.e. after the device fully relaxed to its original state, do not change the state of the device, but elicit significant response when the pulse intervals are kept shorter than 10 ms. The response amplitude increases for shorter pulse intervals, reminiscent of the enhanced sensitivity of biological nociceptors within the relaxation process, and could be further tuned as a function of the stimulus amplitude (Supplementary Note 2, Supplementary Figs. 3 and 5).

Application of voltage shocks (representing injuries of increasing severity) drive STARs to their emergency mode or injured or sensitized state (Mode-2), characterized by reduction of the activation threshold (allodynia) and enhanced current responses (hyperalgesia)[25,26]. To demonstrate the sensitization characteristics of STARs, we recorded current responses after application of high amplitude (2 V, 3 s pulse width) voltage shocks (representing injuries). As indicated in Fig. 2b–e, Supplementary Note 2 and Supplementary Fig. 4, the normal-state responses are initially measured at a $V_{gs} = 1$ V. Noxious stimuli or Injury pulses ($V_{gs} \geq 2$ V) are next applied on the gate terminal of the STARs, after which the sensitized current response curves are once again measured at $V_{gs} = 1$ V. Injured or sensitized STARs exhibit higher output currents (hyperalgesia) and reduced current response thresholds (allodynia), when compared to their normal-state operation as shown in Fig. 2b. Injuries of increased severity (amplitude > 2 V: Fig. 2b, Supplementary Note 2, Supplementary Fig. 4b or pulse width (pw)>3 s: Fig. 2c, Supplementary Note 2, Supplementary Fig. 4c) reduce the pain threshold and elevate the response further, demonstrating the unique sliding threshold behaviour of STARs. For example, when sensitized with +2.25 V noxious stimuli, the STAR's response reaches the noxious threshold ($I_{nox} = 3.3$ mA) within 20 pulses and outputs a maximum value of 4.3 mA at the end of 46 pulses; while +2.5 V noxious stimuli induces activation within 19 pulses and shows higher current responses of up to 5.2 mA. The threshold switching behaviour also enables passive healing with time and active healing with curing pulses of opposite polarity as shown in Fig. 2d, e, Supplementary Note 2 and Supplementary Fig. 4d, e. Figure 2 depicts the peak points of current response of the STAR. The corresponding raw output current spikes are shown in Supplementary Note 2 and Supplementary Fig. 4.

Hence, STARs comprehensively emulate all the signatures of their biological counterparts, thanks to the fatigue-less ion migration-relaxation effects at the electrical double layer interface of the three-terminal memristive device used. Moreover, they do not require precise fabrication of ion reservoirs and shallow defects to ensure good cyclability like their two-terminal diffusive memristor counterparts[27–29]. In comparison, CMOS-based implementation of a single nociceptor would require multiple (at least six) transistors and one capacitor wired together to

implement its adaptability to repeated exposure to noxious stimuli (Supplementary Note 3, Supplementary Fig. 6, Supplementary Table 1).

**Satellite learning modules**. While delocalized STARs pre-process the sensory signal via volatile mathematical threshold functions, learning of more complex phenomena like association and conditioning entails non-volatile synaptic weight updates. Plasticity mediated by activity-dependent strengthening and weakening of synaptic connections forms the basis of learning and memory in the human brain[30]. Analogously in neuromorphic architectures, artificial synapses act as weighted connections between layers of the neural network enabling energy-efficient in-memory computations[31]. In this work, SLMs comprise SWARMs that alter their weights upon arrival of temporally causal and acausal stimuli using STDP; and CMOS spiking neurons (SSNs) that modulate their firing rate as a function of synaptic weight and input stimuli.

**Satellite weight adjusting resistive memories**. We configure thin film memtransistors to operate as gated-memristive switches a.k. a. drift mode to functionally emulate the signal processing of a biological synapse. On persistent application of positive voltage pulses with higher amplitude, additional oxygen vacancies are created in the ultra-thin IWO semiconducting channel, modulating its local electronic structure, and resulting in a non-volatile memory. The accompanying stochiometric transformations monitored through X-ray photoelectron spectroscopy provide critical insights into the underlying oxygen-vacancy generation mechanism and corroborates the comprehensive weight update analyses of the SWARMs (Supplementary Note 4, Supplementary Fig. 7)[32]. Figure 3 shows the characterization of SWARMs, the implemented STDP learning rule compared to biological synapses and higher-order associative learning fusing the information from STARs. We characterized STDP as a function of the temporal window between pre and postsynaptic stimuli. Asynchronous electrical spikes of identical amplitude and duration (representing information like texture of surfaces and objects) induce asymmetric STDP functions in SWARMs (Fig. 3b; Supplementary Note 4, Supplementary Fig. 8). The SWARMs implement an anti-Hebbian[33] form of the STDP where a temporal order of first presynaptic activity followed by postsynaptic activity leads to long-term depression (LTD) while the reverse order leads to long-term potentiation (LTP). This weight modulation in turn modulates the firing rates of neurons, triggering motor responses to avoid potential physical damage, when the texture associated with the noxious stimulus is detected.

To demonstrate the higher-order associative learning in SLMs, we train four of our SWARMs with spikes as illustrated in Fig. 3c. The spikes could be generated using rigid CMOS SSN circuits

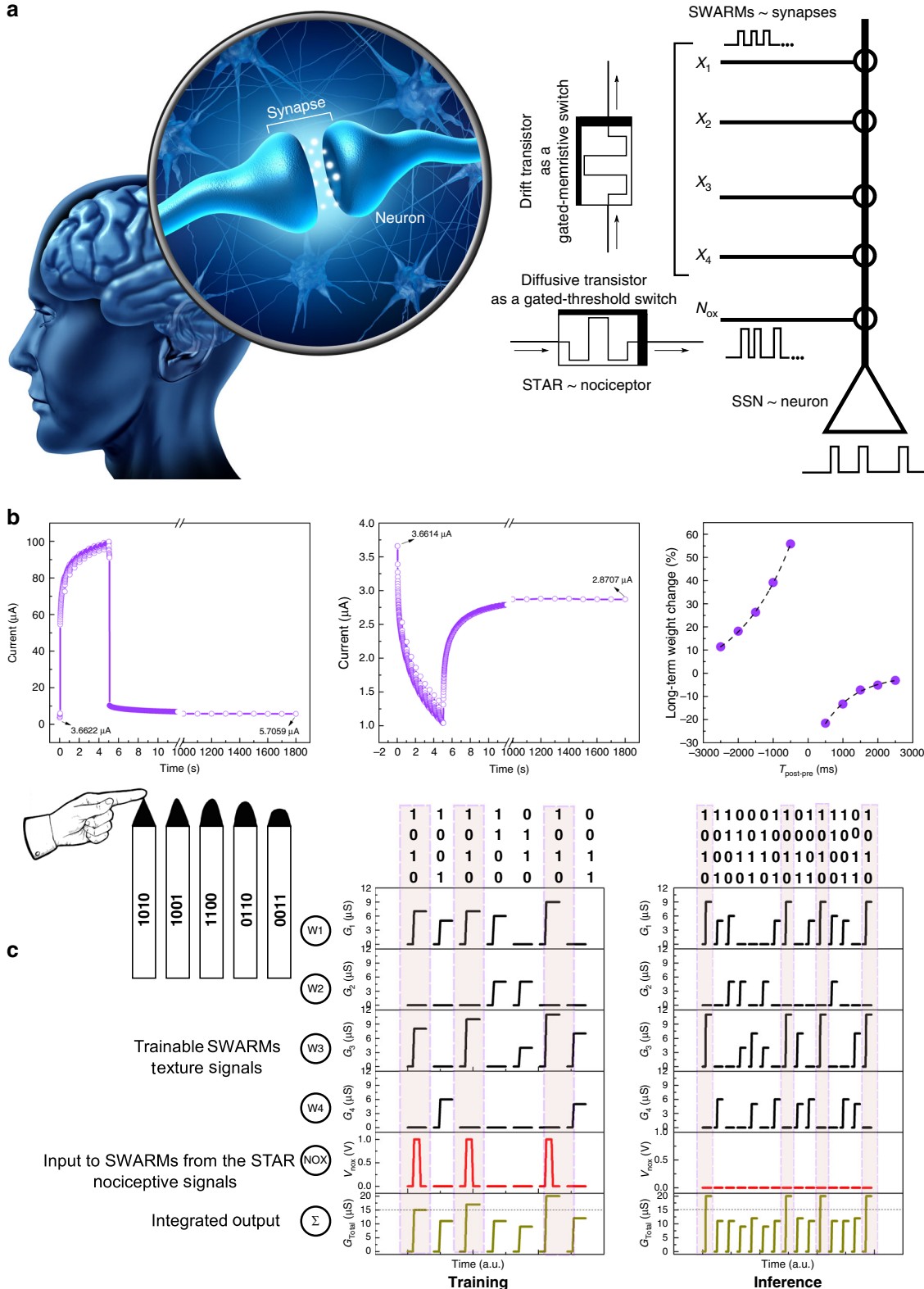

**Fig. 3 Signal processing of satellite learning modules (SLMs).** Associative learning of texture and nociceptive signals. **a** Biological and artificial neural network. Relative timing between pre- and postsynaptic spikes create voltage differences across synapses/satellite weight adjusting resistive memories (SWARMs), tuning the firing rate of neurons/satellite spiking neurons (SSNs). In the proposed approach, teacher signals from the nociceptor/STAR modulates the synaptic weights creating association. **b** Weight changes in the SWARM follow an anti-Hebbian spike-timing-dependent plasticity (STDP) rule. Representative raw I–t curves of long-term potentiation (LTP) and depression (LTD) are shown for clarity. **c** Associative learning of pain and texture signals using satellite threshold adjusting receptors (STARs) and SWARMs. Four SWARMs are trained with signals related to the texture of objects and noxious output signals from STARs.

such as in Supplementary Note 5, Supplementary Figs. 9–11 and Supplementary Table 2 that implement an integrate and fire neuron or could be generated from Mott memristors[17] and flexible ring oscillators that encode stimulus intensity in firing rates. The pattern "1010" corresponds to the signal of interest (texture information corresponding to a very sharp pencil tip) while all other patterns namely "1001", "1100", "0110" and "0011" represent information not correlated with a noxious stimulus (texture information of pencil tips of decreasing sharpness). After three training repetitions, a strong association is developed between the texture of the surface of the object (represented by the patterns) and pain (relayed from STARs), thanks to a significant total weight change (>15 µS) corresponding to the sharp pencil tip ("1010"), while all other inputs fail to cause significant weight changes. The high conductance readouts corresponding to the pattern "1010" during the inference stage even in the absence of the nociceptive signal from STARs demonstrate the fault tolerance enabled by the higher-order associative learning ability of SLMs. Implementation of this associative learning near the sensor node highlights the utility of this decentralized neuromorphic approach in increasing fault tolerance and reducing the wiring complexity for large-area sensing. Implementation of a similar associative logic on a traditional CMOS platform would require multiple interconnected elements with multi-state non-volatile storage capabilities. However, the gated-memristive configuration of SWARMs ensures that one single device can be trained to display this complex association/conditioning rule, minimizing the footprint of the involved circuitry[34].

**Self-healing peripheral sensing and computation.** Self-repair of the components in direct contact with the environment is a crucial capability of biological tissues[35,36] that would support reliable operation of robots in unconstrained environments. To this aim, STARs and SWARMs are designed with self-healable ionic gels/dielectrics that heal themselves when subjected to damage (Fig. 4a). The basic concept in the design of this ion gel is to combine a polar, stretchable polymer with mobile species of the ionic liquid. The ion–dipole interactions—forces between charged ions and polar groups on the polymer increases as the ion charge or molecular polarity increases. Upon addition of high-ionic-strength ionic liquids into the polymer, there are two effects: first, the ionic liquid will plasticize the polymer chains to a much lower glass transition temperature below room temperature; and second the polymer chain diffusion is facilitated by ion–dipole interactions. This allows the polymer to autonomously repair themselves at room temperature. In contrast to majority of the healable systems that depend only on strong intermolecular interactions to glue them back together, requiring manual rejointing of the two cut ends of the polymer pieces[37,38], our physical cut creates a 10 µm gap between the two parts of the polymer film. The reduced glass transition temperature of the ionic liquid–polymer combination allows the polymer to flow back across the 10 µm gap and stitch itself back together with the intermolecular interactions. Thus, we utilize both the reduced glass transition temperature and high intermolecular interactions for healing. Since all electronic devices are typically in a thin film format, this design choice of a material with both the mechanisms of healing is significant for such applications.

Here, the ion gels are composed of a highly polar fluoro-elastomer-poly(vinylidene fluoride-co-hexafluoropropylene) P (VDF-HFP) with very high dipole moment, together with a stable low vapour pressure ionic liquid-1-ethyl-3-methylimidazolium bis(trifluoromethylsulfonyl) imide ([EMI] + [TFSI]− or EMITFSI). The high electronegativity of fluorine, the strong electrostatic nature of the carbon–fluorine (C–F) bonds in the fluoro-elastomer and the fluorine-rich ionic liquid makes the ion gel hydrophobic. Previously, density functional theory calculations have estimated attractive binding energy of ion–dipole interaction, i.e. between a single oligomer of PVDF-HFP and an imidazolium cation to be ~22.4 kcal mol$^{-1}$, nearly twice the binding energy between oligomers (11.3 kcal mol$^{-1}$)[39]. In addition to strong ion–dipole interactions, the $CF_3$ pendant group's steric hindrance provides higher free volume for mobile ions, resulting in higher ionic conductivity and self-healing capability[40]. Additionally, the ionic liquid's high miscibility with the polymer makes the ion gel highly transparent, with an average transmittance of over 87% under visible light. Upon injury, the ionic liquid inclusions trigger the healing process by improving the thermal mobility of the polymer housing via a plasticizing mechanism[41]. The self-healing nature of the proposed ion gels via spectroscopic and mechanical analysis is shown in Supplementary Note 6 and Supplementary Figs. 11–15. Fourier transform infrared spectroscopy (FTIR) studies point to ion–dipole interactions between the polymer and the imidazolium-based ionic liquid (Supplementary Note 6, Supplementary Fig. 12). Differential scanning calorimetry (DSC) reflects the lowering of the glass transition temperature and provides direct evidence of the plasticizing effect (Supplementary Note 6, Supplementary Fig. 13), while Thermogravimetric analysis (TGA) reveals thermal stability of the ion gel above 350 °C (Supplementary Note 6, Supplementary Fig. 14). From the stress–strain curves, softening of the matrix and a decrease in the Young's modulus of the ion gels is observed with higher EMITFSI content. The healed sample depicts a similar slope for the stress–strain curve as the pristine sample, indicating no change in the Young's modulus and mechanical stiffness of the material during the healing process. We describe the mechanical self-healing efficiency as the proportion of restored toughness relative to the original toughness (the area under the stress–strain curve), since this approach considers both stress and strain restoration[40]. The sample shows a healing efficiency of 27% in terms of maximum strain at break (ultimate strain) and an impressive 67% in terms of peak load (Supplementary Note 6, Supplementary Figs. 15 and 16). Figure 4b and Supplementary Note 6 and Supplementary Fig. 17 show the optical and scanning electron microscopy images of the drop casted films at various stages of the damage-heal process at room temperature across 24 h after a 10-µm wide knife cut. The ion–dipole interaction and the plasticizing effect both contributes to the self-healing property of the materials. Figures 4c and 5, Supplementary Note 6 and Supplementary Figs. 18 and 19 show the functional electrical recovery of STARs and SWARMs at various stages of the damage and healing process.

Figure 5a depicts the healing behaviour of short-term plasticity of SWARMs. Paired-pulse facilitation (PPF) refers to a short-term homosynaptic facilitation in which the postsynaptic response to the second action potential is much larger relative to the first due to the accumulation of residual $Ca^{2+}$ in the presynaptic terminal. The degree of facilitation is greatest when the pulse interval is kept shortest, i.e., when the $Ca^{2+}$ ions are not allowed to return to the baseline concentration prior to the second stimulus[42]. Analogous to this, action potentials (+1.5 V, pw = 20 ms) separated by minute pulse intervals (<50 ms) trigger higher excitatory postsynaptic currents in the second presynaptic spike, resulting in PPF indices well above 100%. The SWARMs exhibit the highest PPF index ~181% for a pulse interval of 10 ms. Increasing intervals result in an exponential reduction of the facilitation indices in accordance with the $Ca^{2+}$ residual hypothesis (Fig. 5a, Supplementary Note 6, Supplementary Fig. 18). The exponential decrease of the facilitation indices indicates the temporal dynamics of the ion relaxation mechanism.

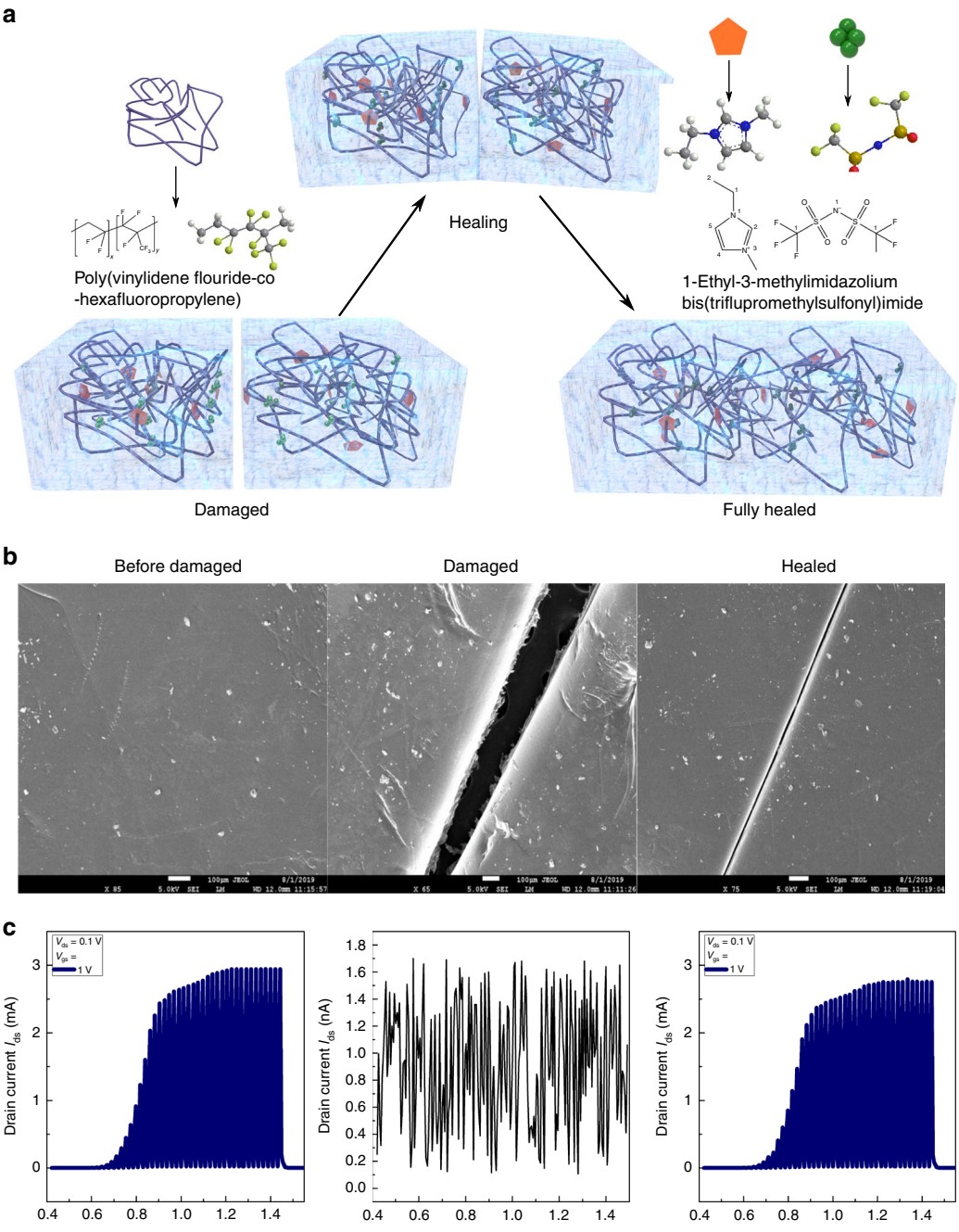

**Fig. 4 Self-healing neuromorphic elements-mechanism and satellite threshold adjusting receptors. a** Our satellite threshold adjusting receptors (STARs) and satellite weight adjusting resistive memories (SWARMs) are designed with self-healable ionic gels/dielectrics that heal themselves when subjected to damage. **b** Upon injury, the ionic liquid inclusions trigger the healing process by improving the thermal mobility of the polymer housing via a plasticizing mechanism (scanning electron microscopy [SEM] images). The corresponding optical images are shown in Supplementary Note 6 and Supplementary Fig. 17. **c** Electrical characterizations recorded on STARs at various stages of the damage and healing process.

Similarly, application of presynaptic pulses of opposite polarity results in a decrease in short-term conductance or paired-pulse depression (PPD). The decay of the depression curves is again exponential, similar to PPF. The devices are then damaged with a knife cut that creates a 10 μm gap. Analysis of the PPF and PPD indices after the healing process indicate good recovery of the conductance levels and restoration of the ion accumulation-relaxation mechanism (Fig. 5a, Supplementary Note 6, Supplementary Fig. 18).

Since SWARMs are utilized to implement associative learning via non-volatile weight changes, we next focus on measurements

of their long-term plasticity behaviour. Supplementary Note 6 and Supplementary Fig. 19 show the representative LTP and LTD curves before damage and after the healing process, and Fig. 5b, c depicts the STDP and LTP-LTD behaviour of our SWARMs as a function of the number of training cycles, before damage and after the healing process, respectively. The weight update trace follows a similar trend before damage and after healing, indicating complete functional recovery and healing of the ion gel dielectric after mechanical damage. The device-to-device variations are captured by the error plots in Fig. 5c. In general, the LTP and LTD weight updates do indicate higher variations after

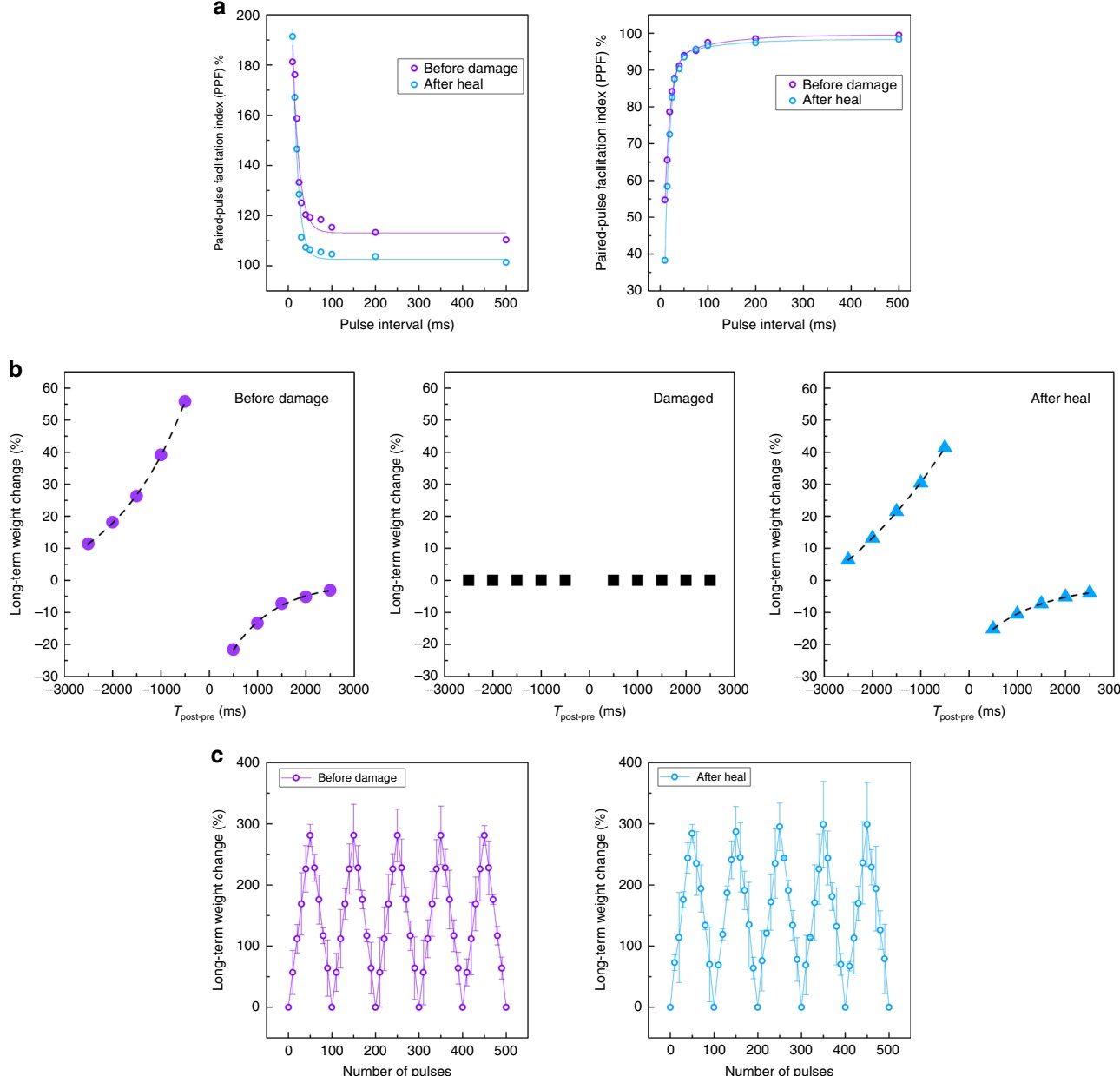

**Fig. 5 Self-healing neuromorphic elements—satellite weight adjusting resistive memories (SWARMs). a** Short-term plasticity. A pair of presynaptic action potentials (+1.5 V, pulse width = 20 ms, interval = 10 ms) triggers a pair of excitatory postsynaptic currents (EPSCs) with increasing amplitude. This phenomenon known as paired-pulse facilitation (PPF) reflects the number of residual carriers during the ion migration-relaxation kinetics (left). Reversal of polarity of the presynaptic action potentials (−1.5 V) result in paired-pulse depression (PPD) with the indices dependent on pulse width and interval of the presynaptic action potentials, similar to facilitation (right). PPF/D indices, defined as [PPF/D = $\left(\frac{A_2}{A_1}\right) \times 100\%$] is plotted as a function of inter-spike interval to demonstrate the decay process. **b** Long-term plasticity. Electrical characterizations of spike-timing-dependent plasticity (STDP) recorded on SWARMs at various stages of the damage and healing process. **c** Controlled long-term potentiation (LTP) and depression (LTD) achieved in SWARMs over 500 switching transitions by applying a series of potentiating (+1.5 V) and depressing (−1.5 V) presynaptic spikes. Each programming/erasing step consists of 10 spikes of pulse width 500 ms. The figure represents the cycle-to-cycle variations during programming and erasing. The error bars capture the device-to-device variations obtained from 20 devices. The LTP and LTD weight updates do indicate higher variations after damage, but the trend of the overall the weight update traces remain consistent even after severe mechanical damage to the ion gel dielectric.

damage, but the trend of the overall weight update traces remain consistent even after severe mechanical damage to the ion gel dielectric. From a yield perspective, 19 out of the 20 damaged samples recover functionally after 24 h of healing time. In comparison, conventional CMOS-compatible dielectrics like $SiO_2$ fail upon mechanical damage as shown in Supplementary Note 6 and Supplementary Fig. 20. The healing behaviour of our devices is further validated in real time by the demonstrations shown in

Supplementary Movies 1–3, Fig. 6, Supplementary Note 6 and Supplementary Fig. 21. The devices heal back after damage and are able to trigger motor responses in the robot.

**Application benchmark**. As a validation step of the proposed sensory signal processing artificial skin, we implemented pain-reflex movements in a robotic arm upon sensing of noxious mechanical and thermal stimuli (Fig. 6, Supplementary Note 6,

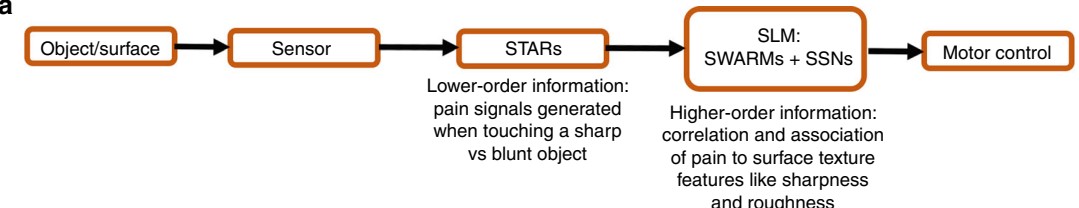

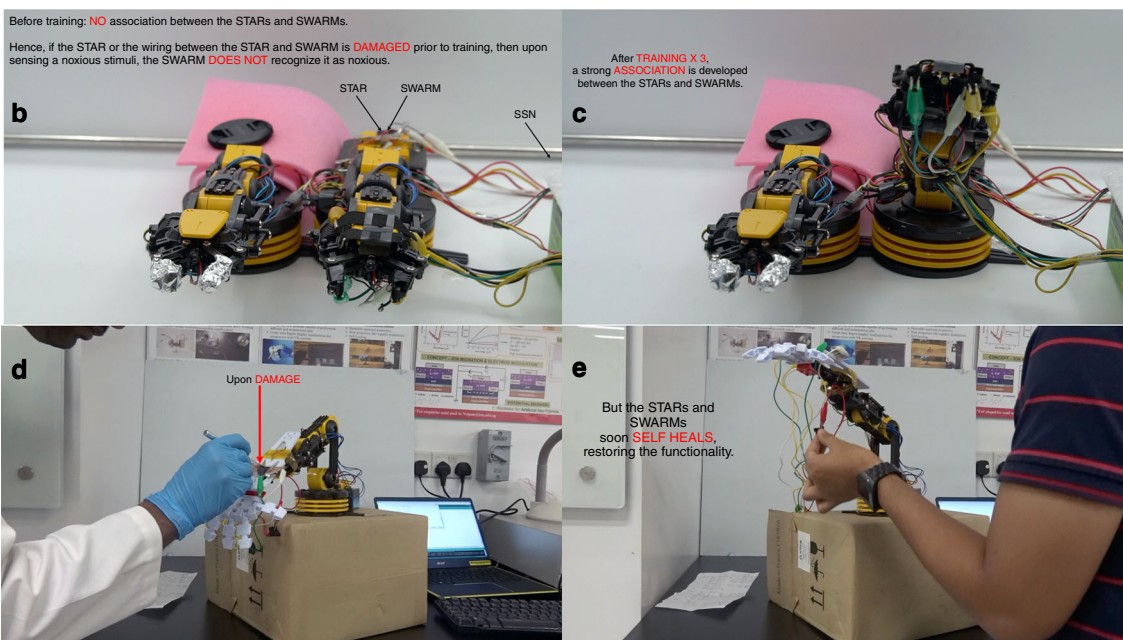

**Fig. 6 Demonstration of the working of decentralized memristive neuromorphic elements for robotics. a** Flowchart of the implemented logic. **b, c** Sensorimotor platform capable of detecting and associating noxious stimuli. Resistance changes to the pressure sensing element triggers generation of warning signals from satellite threshold adjusting receptors (STARs) above a pre-set pain threshold. This in turn changes the weight plasticity of associated satellite weight adjusting resistive memories (SWARMs) in the satellite learning module (SLM), triggering motor responses in the robotic arm. Upon training, an effective association is developed between the STARs and SWARMs enhancing the fault tolerance of this approach. **d–e** Functional recovery of the system upon mechanical damage. Upon damage, the STARs and SWARMs self-heals restoring the circuit functionality. Please refer to Supplementary Movies 1–3 for more details. An expanded version of this figure with more details is provided in Supplementary Note 6 and Supplementary Fig. 21.

Supplementary Fig. 21) that after a training phase are associated to the noxious stimuli. The output pain signals from an array of 16 STARs are sent to 16 SWARMs via an Arduino controller, which acts as the SSN in the experimental setup. On encountering noxious stimuli, the warning outputs from STARs act as punishment signals to SWARMs, modulating their response as shown in Fig. 3c. The internal re-adjustment of the weights in turn modulates the firing rates of SSNs, triggering escape motor responses of a robotic arm to avoid potential physical damage. Supplementary Movie 1 illustrates how associative learning evolves over time between the STARs and SWARMs and how this association enables fault tolerance (identification of noxious signal even after nociceptor damage—in alignment with the findings shown in Fig. 3). Supplementary Movie 2 illustrates how the system as a whole survives and responds to unintentional mechanical damage. Self-healability of the ion gels helps restore functionality of the STARs and SWARMs upon inadvertent mechanical damage. Finally, Supplementary Movie 3 shows the potential of scaling up this concept to fuse more number of sensory inputs and learning modules for better decision-making.

## Discussion

The unique sliding threshold feature of STARs improves signal integrity by filtering lower-order sensory information like pain signals at the location of the sensing nodes. Equipping the sensor

nodes with associative learning capabilities addresses the scalability and data transfer, and helps to reduce the complexity of wiring. While synaptic learning is typically relegated to the central brain in other implementations, we show for the first time how incorporating associative learning via plasticity of synapses (SWARMs) in the robotic equivalent of peripheral nervous system enables fault tolerance (identification of noxious signal even after nociceptor damage). In contrast to very recent artificial afferent nerve implementations with organic synaptic transistor[21] and Mott memristors[20], the proposed system adds learning and decision-making for the first time at the sensor node.

We report the first three-terminal artificial nociceptor (STAR) fully integrated with artificial synapses and neurons, to the best of our knowledge. With respect to existing two-terminal memristive implementations of artificial nociceptors[27–29] which require precise fabrication of ion reservoirs and shallow defects to ensure good cyclability, our STARs seamlessly work on the basis of the ion migration-relaxation effects at the electrical double layer interface without fatigue. From a device perspective, the thin film transistor-based configuration of STARs, SWARMs and SSNs enables emulation of nociceptors, synapses and neurons with the minimum possible number of devices, reducing circuit and wiring complexity. Compared to CMOS implementations, many of the required layers can be facilely and cheaply printed and support bending and stretching, paving the way for the implementation of

flexible neuromorphic robotic skin. Moreover, since the switching mechanism between the high- and low-resistance states depends only on the accumulation and depletion of carriers in the semiconducting channel, both STARs and SWARMs do not require a forming step nor a compliance current control to avoid damage, enhancing their cyclability and simplifying the peripheral circuitry design. In addition, the gating strategy also allows access to a large number of states when compared to traditional two-terminal memristors[34]. Since the memtransistor can be configured to operate as both a nociceptor and synapse, this allows us to facilely build a platform with a single core device configuration unlike conventional two-terminal memristors that require special design of diffusive and drift configurations.

This is the first report of healable neuromorphic memristive devices—artificial nociceptors and synapses to the best of our knowledge. Although sensorized artificial synapses approaching human skin-like performance in terms of mechanical sensing and form factor have been very recently demonstrated[21,43], the ability to repeatedly self-heal neuromorphic circuit elements has not been demonstrated yet. Such repeatable electrical and mechanical healing at room temperature (even at the same damage location) is a breakthrough towards the deployment of robots and prostheses in real-world applications. Compared to the very recent report on electronic skin where healing is limited to only the substrate[10], we demonstrate complete functional and mechanical self-healing of all devices and the associative learning within the learning modules enables good signal integrity even if the nociceptor is damaged after learning, enhancing fault tolerance.

In summary, the proposed concept of decentralized intelligence finds close correlations to very recent biological investigations that prove important functions like pain reflex and motor control to be implemented at the level of the spinal cord[44]. This approach can be readily extended to other sensing modalities and material platforms. While still at a prototypical stage, this works lays down a novel framework for building a memristive robotic nervous system with direct implications for intelligent robotics and prostheses[45]. The self-healing capability of these intelligent devices opens up the possibility that robots may one day have an artificial nervous system that can repair itself. This ability is hitherto not demonstrated for hardware neuromorphic circuits and is timely especially with the future of electronics and robotics going soft.

## Methods

**Solid-state ionic dielectric**. The ionic liquid [EMI][TFSI] was initially dried in vacuum for 24 h at a temperature of 70 °C. Next, P(VDF-HFP) and [EMI][TFSI] were co-dissolved in acetone with a weight ratio of 1:4:7. The ion gels were further dried in vacuum at 70 °C for 24 h to remove the residual solvent, after which it was cut with a razor blade, and then laminated onto the substrate of choice. To determine the nature of interaction between the ionic liquid and polymer matrix, FTIR spectroscopy was performed using FTIR spectrum GX, PerkinElmer. The plasticizing effect was investigated using DSC (DSC TA Instruments 2010) at a ramping rate of 10 °C min$^{-1}$. The samples were tightly sealed in aluminium pans, and the measurements were carried out while heating up the sample to 200 °C, followed by cooling down to –80 °C, at a heating and cooling rate of 10 °C min$^{-1}$. The degradation (working) temperature of the ion gel was measured by TGA (TGA-Q500). The self-healing nature of the ion gels was observed and captured under a polarizing optical microscope (Olympus, CX31-P).

**Device fabrication and characterization**. IWO thin films (thickness ~7 nm) were deposited on SiO$_2$/Si wafers at room temperature using an RF magnetron sputtering technique with an In$_2$O$_3$:WO$_3$ (a-IWO) (98:2 wt%) target at a gas mixing ratio of Ar:O$_2$ (20:1), total chamber pressure of 5 mtorr and RF power of 50 W. ITO source and drain contacts (thickness ~100 nm) were then sputter deposited through a shadow mask using an In$_2$O$_3$:SnO$_2$ (90:10 wt%) target. The devices were then annealed at 200 °C for 30 min in ambient environment for optimized transistor performance. Ion gels were next laminated on to these devices. Contacts to the ion gel were made directly via a side metal gate (Ag) or directly using the probe station tip. Electrical measurements were carried out in a Desert Cryogenics (Lakeshore) probe station using Keithley 4200-SCS semiconductor characterization

system. Capacitance measurements were carried out using an Alpha A Analyzer, Novocontrol analyser, over a frequency range of 1 Hz to 10 kHz.

## Data availability

The data that support the findings of this study are available from the corresponding author upon reasonable request.

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

## Acknowledgements

The authors would like to acknowledge the funding from MOE Tier 1 grants: RG87/16, RG 166/16 and MOE Tier 2 grants MOE2016-T2-1-100 and MOE2018-T2-2-083. We thank Prof. Koh Soo Jin Adrian (Department of Mechanical Engineering, NUS) for access to dielectric spectroscopy measurements.

## Author contributions

R.A.J., A.B. and N.M. conceived the experiments. R.A.J. performed all the optoelectronic characterizations under the supervision of N.M. and A.B. N.T. synthesized the healable ionic dielectric and performed all mechanical characterizations together with Ankit. M.I.B.P. and C.J.Y. programmed the arduino controller and helped set up the final demonstration with guidance from M.R.K. J.B. designed the neuron circuit and S.K.B simulated CMOS version of the STAR under the supervision of A.B. N.T. fabricated the IWO transistors. A.N. and S.K.V. helped formulate the figures and reviewed the manuscript. R.A.J., C.B., A.B. and N.M. wrote the manuscript with comments from all authors.

## Competing interests

The authors declare no competing interests.
