## [Peer Review File · Nature Communications]

Reviewers' Comments:

Reviewer #2:

Remarks to the Author:

The authors reported a few devices and their characteristic behavior, namely an ion-gated transistor as a nociceptor, a 'self-healing' polymeric materials with ion gel for synapses and a 'satellite spiking' neuron composed of two op amps, a capacitor, a few resistors and a memristor. Although they were trying to integrate all three modules into a comprehensive memristive system, the effort is not necessarily coherent in my opinion. As a matter of fact, both the nociceptor and the neuron have some issues that comprise the novelty of this work. For example, the authors claimed that the nociceptor exhibits 'allodynia' but this claim is not substantiated with explicit experimental data. By definition, allodynia means a 'damaged' device can respond to a stimulus that normally does not lead to such a response. In order to demonstrate allodynia in the current device, the current response of the device should be measured under a voltage pulse below the threshold after different degrees of 'damages'. The integrate and fire behavior of the 'satellite spiking neuron', on the other hand, is a result of the capacitor and the op amps, not intrinsic to the memristor itself. It is hence a stretch to call this a memristive neuron.

That being said, I do find the work on the self-healing synapses interesting. Although there have been quite a few self-healable materials developed in the past, there is few using such material systems for neuromorphic computing related applications. It is hence suggested that the authors focus on this part of the work, thoroughly study the mechanism underpinning the self-healing phenomenon, and extensively study the synaptic behavior in more details (not just limit to the STDP). The studies on the mechanical properties, as shown in the SI, are sufficient. But these studies are not a differentiator from other published work on self-healing systems.

Reviewer #3:

Remarks to the Author:

In the present manuscript "Self-Healable Memristive Neuromorphic Elements for Decentralized Sensory Signal Processing in Robotics" the authors present a sensor system exploiting memristive devices in neuromorphic nociceptors. The use a three-terminal memristive device consisting of a $\text{In}_2\text{O}_3:\text{WO}_3$ channel that is controlled via an gated ionic liquid. This device is used in a volatile or non-volatile manner in a spiking neuron circuit as "SWARM" device and as "STAR" device. Based on these three components a sensor system is build and attached to a robot as nociceptor. The study is quite interesting and shows novel results. It is, however, not clear what the main achievement is. According to the title the device is novel but it is not really described in the manuscript. Only with the main manuscript it is even not clear that the same device type is used for the different components of the sensory systems. The manuscript needs to be revised in order to emphasize the novelty of the approach (including a comparison to literature). In addition, the device should be described in the main text and not only the supplement. Besides these major concerns, the following comments need to be addressed in a revised version.

- 1) The fonts in the figures are too small and should be enlarged.
- 2) In Fig. S1c, it is no clear why 0.2 V is a threshold. The I-V trace looks rather gradual. Moreover, the sweep rate is missing as well.
- 3) The switching mechanism of the three-terminal device is not clear. Which ions are moving? Why does the ion movement leads to a resistance change (volatile and non-volatile). Is there any evidence for the switching mechanism based on ionic movement?
- 4) The authors should add a device characterization for the non-volatile switching behavior in the supplement. What is about statistics, device-to-device variability, cycle-to-cycle variability, and endurance data?
- 5) Please provide more statistics for the trends shown in Fig. S2. For 1.1 V and 1.2 V only 1 pulse seems to be shown. What is the delay between the pulses?
- 6) In Fig. S2, the authors show first data in the mA range. The relaxation behavior in Fig. S3

shows data in the nA range. How can the system adapt to 6 orders of magnitude different current response? Why is this behavior called relaxation behavior? The current seems to drop by 6 order of magnitude immediately.

7) The experiment of Fig. S3 is not clear. The authors claim to use a voltage $V < V_{th}$. Here, +0.5 V are used but before a threshold voltage of -0.2V was mentioned.

8) The proposed device is a 3-terminal device; a kind of synaptic transistor. The introduction is a little bit confusing as memristive devices are commonly liked to passive 2-terminal device. Thus, the authors should clearly state from the beginning that a 3-terminal device is used.

9) The authors mention a drift mode and a diffusion mode of the device. Both modes should be introduced before. Drift and diffusion mode are not common knowledge.

10) It is not clear how the voltage is applied to the three-terminal device in the experiments in Fig. S5.

11) The device stack should be explained in more detail. It is not clear for example how the gate contact to the ionic liquid is realized.

12) In Fig. S6 R_{mem} is shown as a two-terminal device, but is supposed to be a three-terminal one. Is this a different device as the proposed one?

REVIEWER COMMENTS

Reviewer #2 (Remarks to the Author):

The authors reported a few devices and their characteristic behavior, namely an ion-gated transistor as a nociceptor, a ‘self-healing’ polymeric materials with ion gel for synapses and a ‘satellite spiking’ neuron composed of two op amps, a capacitor, a few resistors and a memristor. Although they were trying to integrate all three modules into a comprehensive memristive system, the effort is not necessarily coherent in my opinion. As a matter of fact, both the nociceptor and the neuron have some issues that comprise the novelty of this work. For example, the authors claimed that the nociceptor exhibits ‘allodynia’ but this claim is not substantiated with explicit experimental data. By definition, allodynia means a ‘damaged’ device can respond to a stimulus that normally does not lead to such a response. In order to demonstrate allodynia in the current device, the current response of the device should be measured under a voltage pulse below the threshold after different degrees of ‘damages’. The integrate and fire behavior of the ‘satellite spiking neuron’, on the other hand, is a result of the capacitor and the op amps, not intrinsic to the memristor itself. It is hence a stretch to call this a memristive neuron.

That being said, I do find the work on the self-healing synapses interesting. Although there have been quite a few self-healable materials developed in the past, there is few using such material systems for neuromorphic computing related applications. It is hence suggested that the authors focus on this part of the work, thoroughly study the mechanism underpinning the self-healing phenomenon, and extensively study the synaptic behavior in more details (not just limit to the STDP). The studies on the mechanical properties, as shown in the SI, are sufficient. But these studies are not a differentiator from other published work on self-healing systems.

We thank the referee for the positive comments on our work and acknowledging the significance of our work on self-healing memristive and neuromorphic devices. We also sincerely appreciate the reviewer’s suggestions to improve our manuscript by making more objective statements.

“Although they were trying to integrate all three modules into a comprehensive memristive system, the effort is not necessarily coherent in my opinion.”

We thank the reviewer for the comment. We would like to reiterate the vision of our approach, most significant findings and outcomes of our work to provide a more cohesive picture. In addition to the first demonstration of self-healable synaptic transistors fashioned as artificial nociceptors and synapses, the below sections provide an outline of the novelties.

Overview:

We envision a decentralized approach to sensory information processing with intelligence shifted to the location of the sensing nodes as a novel neuromorphic approach to tackle the issues of information processing and learning in robotic skins. In comparison to the existing centralized approaches that are disadvantaged by the need to constantly shuttle large amounts of data from the periphery to the center, with wiring, latency, fault tolerance and robustness issues; we decentralize learning and embed decision making at the sensor node by fusing sensing elements with neuromorphic devices-namely nociceptors (STARs), synapses (SWARMs) and neurons (SSNs).

State-of-the-art:

The concept of decentralized signal processing is considered as a highly promising solution to manage information processing in robotic skins in the robotics community [Cheng, G. et al. 2019. *Proceedings of the IEEE*, 107(10), pp.2034-2051.]. State-of-the-art approaches include distributing embedded sensors and system-on-chip processors across the robot's body and utilisation of spike-generating schemes for efficient addressing (Eg: Address Event Representation [Bartolozzi, C. et al. 2017 *IEEE/RSJ International Conference on Intelligent Robots and Systems (IROS)* 166–173] and spread spectrum techniques [Lee, W. W. et al. 2019 *Sci. Robot.* 4, eaax2198.] for information processing. However, the processor-based implementations have difficulties matching the scale, connectivity and energy efficiency of their biological counterpart and often place little emphasis on fault tolerance against mechanical damage, essential for robots working in uncertain environments. Most critically, from the device perspective, such complementary metal-oxide-semiconductor (CMOS)-based approaches do not suffice the scalability requirements to finally match the connectivity of the human nervous system. For example, implementation of a single CMOS synapse portraying spike-timing dependent plasticity (STDP)-based learning rules require multiple transistors, resistors and capacitors, hindering its scalability.

Our Novel Approach:

We present a novel neuromorphic approach based on the unique sliding threshold behaviour of biological nociceptors and associative learning in synapses to shift intelligence to the location of the sensor nodes, resulting in decentralized decision making. STARs pre-process the sensory signals via volatile mathematical threshold functions such as relaxation and sensitization, while learning of more complex phenomena like association and conditioning are carried out by SWARMs and SSNs via non-volatile synaptic weight updates. This approach lowers the temporal redundancy of event-based sensory signals and reduces the amount of data shuttled to the central processing system, hence lowering the latency and wiring demands. While synaptic learning is typically relegated to the central brain in other implementations, we show for the first time how incorporating associative learning via plasticity of synapses (SWARMs) in the robotic equivalent of peripheral nervous system enables fault-tolerance (identification of noxious signal even after nociceptor damage).

The memristor-based implementation allows us to facilely emulate the electrical signatures and weight update rules of nociceptors, synapses and neurons with simple circuitry. While artificial synapses have been realized widely, a comprehensive memristive implementation of neuromorphic sensory receptors and their fusion with memristive synapses and neurons have hitherto not been demonstrated. In comparison to the artificial afferent nerve implementations utilizing oscillators, Mott memristors [Nature Communications, 2020, 11(1), pp.1-9.] and organic synaptic transistors [Science, 2018, 360(6392), pp.998-1003.] with physically separate signal transduction and processing, we propose a novel decentralized scheme and demonstrate decision making at the sensor node as a viable solution to address the peripheral sensory signal processing in robotics.

Novelty at the device/material level:

1. *We report the first three-terminal artificial nociceptor.* Previous reports on artificial nociceptor were based on diffusive 2-terminal memristors [Yoon, J.H. et al. 2018. *Nature Communications*, 9(1), pp.1-9.; Kim, Y. et al. 2018. *Advanced Materials*, 30(8), p.1704320.].

2. While artificial synapses have been realized widely, a comprehensive memristive implementation of neuromorphic sensory receptors and their fusion with memristive synapses

and neurons have hitherto not been demonstrated. In comparison to the artificial afferent nerve implementations utilizing oscillators, Mott memristors [Nature Communications, 2020, 11(1), pp.1-9.] and organic synaptic transistors [Science, 2018, 360(6392), pp.998-1003.] with physically separate signal transduction and processing, we propose a novel decentralized scheme and demonstrate decision making at the sensor node as a viable solution to address the peripheral sensory signal processing in robotics.

Flow of the manuscript:

1. Systematic analysis of artificial nociceptors-STARs.
2. Detailed analysis of artificial synapses-SWARMs.
3. Integration of SWARMs with a CMOS neuron to demonstrate neuronal spiking.
4. System-level integration of 1-3. Real-time demonstration on a prototypical robot hand.

We have now amended the manuscript to highlight the novelties demonstrated. Please refer to pages 3, 16, 19-20 in the main text. In summary, we believe that the systematic analysis and proof-of-concept experiments we demonstrate goes beyond the state-of-the-art in memristive technology and their demonstrations in signal processing in robotics, and will be of interest to a broad scientific audience in materials science, computing, applied physics and electrical engineering.

“For example, the authors claimed that the nociceptor exhibits ‘allodynia’..... should be measured under a voltage pulse below the threshold after different degrees of ‘damages’.”

We thank the reviewer for the comment. We would like to clarify that the data presented earlier in Figures 2B-E indeed depicted current responses of STARs when measured with voltage pulses below the threshold after different degrees of ‘damages’, in line with the accurate definition of allodynia. This was explicitly mentioned in the earlier version as *“In this work, voltage triggers of weak amplitude generate normal-state responses, while intense voltage shocks of high amplitude represent injury and generate sensitized responses..... To demonstrate the “sensitization” characteristics of STARs, we recorded current responses after application of high-amplitude (2V, 3s pulse width) voltage shocks (representing injuries). ”*

However based on the reviewer comment, we notice that the legends provided in the previous version of the figures on the pulsed voltages can be misleading and might be the source of confusion. All the data collected for Figure 2 follow the below methodology:

1. The normal-state responses are initially measured at a $V_{gs}=1V$ (voltage pulses below the threshold).
2. Noxious stimuli or Injury pulses ($V_{gs}\geq 2V$) are next applied on the gate terminal of the STARs.
3. The sensitized current response curves are once again measured at $V_{gs}=1V$.

We have now explicitly explained the methodology of the experiments in the main text and supporting information and have amended the legends and figure captions to indicate this clearly. We believe this clarifies the definition of allodynic and hyperalgesic responses. Please refer to the caption of Figure 2 and page 8 (main text) and Figure S4 and page 10 (supporting information).

We also recognize that our earlier illustration of pain threshold current level that is reached in the emergency state upon application of injury pulses and the normal state upon extensive application of pulses can be misinterpreted. Hence, we have now carefully redefined our pain threshold current level as 3.3mA (as opposed to 2.9mA in the earlier version), utilizing the distinct response of our device in their normal and sensitized modes of operation. This new threshold is not reached in the devices' normal state response even after extensive pulsing. We would like to point out that this change of pain threshold from 2.9mA to 3.3mA does not alter the definition of allodynic and hyperalgesic responses in any manner, but imparts better clarity to the readability of the figure since the pain threshold is not reachable in the normal-state even after extensive voltage pulsing. This delineates the responses in the normal and emergency states more clearly. We have indicated this new reference pain threshold current level in the Figures R1 and R2 (shown below).

Based on the reviewer 3's suggestion, we have also split the earlier Figure 2 now into the main text and supporting information so as to accommodate larger sized fonts in the figures for better clarity. We now provide a schematic of the input pulsing scheme in the supporting information Figure-R2 below. Please refer to the new Figure 2 in main text and Figure-S4 in the supporting information in this revised version. We once again thank the reviewer for pointing this out. It has improved the clarity of both our figure and explanation.

Figure R1. Nociceptive Signal Processing in STARS. Pulsed response of STAR. Response of the device to a train of input voltage pulses of width 16ms and amplitude 1V. Voltage pulses of weak amplitude ($V_{gs}=1V$) are

used to generate normal-state responses, while intense voltage shocks of high amplitude ($V_{gs} \geq 2V$) representing injury generate sensitized responses. Normal-state responses are initially measured at a $V_{gs} = 1V$. Noxious stimuli or Injury pulses ($V_{gs} \geq 2V$) are next applied on the gate terminal of the STARs, after which the sensitized current response curves are once again measured at $V_{gs} = 1V$. Increased amplitude (A) and pulse width (B) of the noxious stimuli enhance the current response (“hyperalgesia”) and reduce the incubation time/threshold (“allodynia”), akin to biological nociceptors. (A) The device is subjected to injuries represented by voltage shocks of 2V, 2.25V and 2.5V, pulse width of 3s. The sensitized responses are measured post-injury at $V_{gs} = 1V$. (B) The device is subjected to injuries represented by voltage shocks of 2V, pulse width of 3, 5 and 7s. The sensitized responses are measured post-injury at $V_{gs} = 1V$. The threshold switching behaviour also enables (C) “passive healing” with time and (D) “active healing” with “curing” pulses of opposite polarity. For (C), the device is subjected to an injury of 2.5V, pulse width of 3s. The responses are measured post-injury at $V_{gs} = 1V$ after waiting for 5, 10 and 15 minutes respectively. For (D), the device is subjected to an injury of 2.5V, pulse width of 3s. Next, active healing pulses (AHP) of -1.5V, -2V and -2.5V are applied for 5s. The responses are next measured post-healing at $V_{gs} = 1V$. The graphs depict the peak points of current response of the STAR. The corresponding raw output current spikes are shown in Figure-R2.

Figure R2. Nociceptive Signal Processing in STARs. Raw output current response corresponding to Figure R1. The spike waveform shown on top represents the inputs applied on the STARs to generate the normal and

sensitized responses. Increased amplitude (A) and pulse width (B) of the noxious stimuli enhance the current response (“hyperalgesia”) and reduce the incubation time/threshold (“allodynia”), akin to biological nociceptors. The threshold switching behaviour also enables (C) “passive healing” with time and (D) “active healing” with “curing” pulses of opposite polarity.

“The integrate and fire behavior of the ‘satellite spiking neuron’.... a stretch to call this a memristive neuron.”

We thank the reviewer for the comment. We agree with the reviewer that the neuronal spiking dynamics we present in this work depend on the charging and discharging of the RC circuit and only utilizes the memristive property of the SWARM as a synaptic weight. It was not our intention to call the SSN a memristive neuron, since this would require memristors with 2nd order effects [Kumar, S. et al. 2017. *Nature*, 548(7667), pp.318-321.]. For better clarity, we now call our neuron a CMOS neuron. Please refer to pages 4 and 12 in the main text and Note-5 Supporting Information.

The intention of integrating the SWARMS with off-the-shelf resistors, capacitors and op-amps to form a SSN circuit was to show the compatibility of STARS and SWARMS with existing CMOS-based neuron implementations and more importantly, to emulate all three basic fundamental building blocks of a peripheral learning system-namely, artificial nociceptors, synapses and neurons. This also forms an integral part of our robot demonstration, shown in Supplementary Movies-V1-3.

“the work on the self-healing synapses interesting.extensively study the synaptic behavior in more details (not just limit to the STDP).”

We thank the reviewer for the comment. The basic concept in the design of this ion gel is to combine a polar, stretchable polymer with mobile species of the ionic liquid. The ion–dipole interactions- the forces between charged ions and polar groups on the polymer increases as the ion charge or molecular polarity increases. Upon addition of high-ionic-strength ionic liquids into the polymer, there will be two effects: first, the ionic liquid will plasticize the polymer chains to a much lower glass transition temperature below room temperature; and second the polymer chain diffusion is facilitated by ion–dipole interactions. This allows the polymer to autonomously repair themselves at room temperature. Previously DFT calculations have verified the enhanced intermolecular stability and the favourable ion–dipole interactions within the self-healing polymers. The estimated attractive binding energy of ion–dipole interaction, i.e. between a single oligomer of PVDF-HFP and an imidazolium cation is ~22.4kcal/mol, nearly twice the binding energy between oligomers (11.3kcal/mol) [Cao, Y. et al. 2017. *Advanced Materials*, 29(10), p.1605099.]. Majority of the healable systems depend only on strong intermolecular interactions to glue them back together. Thus in typical healing demonstrations, the 2 cut ends of the polymer pieces are manually forced back together for the intermolecular interactions to be triggered [Colquhoun, H. 2012. *Nature Chemistry* 4, 435–436., Ahn, B.K. et al. 2014. *Nature Materials*, 13(9), pp.867-872.]. In contrast, our physical cut creates a 10µm gap between the 2 parts of the polymer film. The reduced glass transition temperature of our ionic liquid-polymer combination allows the polymer to flow back across the 10µm gap and “stitch” itself back together with the intermolecular interactions. Thus, we utilize both the low reduced glass transition temperature and high intermolecular interactions for healing. Since all electronic devices are typically in a thin film format, we believe our design choice of a material with both the mechanisms of healing is significant for such applications.

The self-healing nature of the ion gels in this work is investigated via a combination of Fourier transform infrared (FTIR) spectroscopy, differential scanning calorimetry (DSC) and

thermogravimetric analysis (TGA). FTIR studies indicate ion–dipole interactions between the polymer and the imidazolium-based ionic liquid Figure-S12. Differential scanning calorimetry (DSC) reflects the lowering of the glass transition temperature and provide direct evidence of the plasticizing effect (Figure-S13), while Thermogravimetric analysis (TGA) reveals thermal stability of the ion gel above 350 °C (Figure-S14). Softening of the matrix and decrease in the Young’s modulus of the ionic gels with higher EMITFSI content is evident from the stress-strain curves. It is worthwhile to mention that the healed sample has similar slope for the stress-strain curve as the pristine sample, indicating no change in the Young’s modulus and mechanical stiffness of the material during the healing process. We describe the mechanical self-healing efficiency as the proportion of restored toughness relative to the original toughness (the area under the stress–strain curve), since this approach considers both stress and strain restoration. [Cao, Y. et al. 2019. *Nature Electronics*, 2(2), pp.75-82.]. The sample shows a healing efficiency of 27% in terms of maximum strain at break (ultimate strain) and an impressive 67% in terms of peak load (Figures-S15,16). We have now expanded this discussion now in the main text. Please refer to pages 14-16.

More importantly, as per the reviewer’s suggestion, we have now carried out extensive studies on the synaptic behaviour of SWARMS before and after healing. In addition to the STDP data previously presented, here we first study paired-pulse facilitation and depression, forms of short-term plasticity. To analyse non-volatile weight updates that underlie associative learning, we next investigate long-term plasticity systematically. We map out weight updates as a function of both time-stepped inputs (LTP-LTD curve) and synchronous inputs (STDP). From the device perspective, self-healing of the ion gel restores the electrical double layer gate capacitance of our memtransistor, in turn re-establishing the ion migration relaxation dynamics at the semiconductor-dielectric interface, responsible for the plasticity in STARS and SWARMS.

Figure R3. Short-term Plasticity in SWARMS. A pair of presynaptic action potentials (+1.5V, pulse width = 20ms, interval = 10ms) trigger a pair of excitatory postsynaptic currents (EPSCs) with increasing amplitude. This

phenomenon known as PPF reflects the number of residual carriers during ion migration relaxation kinetics in the ionic-gated mode. Reversal of polarity of the presynaptic action potentials (-1.5V) result in PPD with the indices dependent on pulse width and interval of the presynaptic action potentials, similar to facilitation. (A) shows representative plots of the drain current before damage and after the healing process for both PPF and PPD. (B) PPF/D indices, defined as $[PPF/D = \left(\frac{A_2}{A_1}\right) * 100\%]$ is plotted as a function of inter-spike interval to demonstrate the decay process.

We first investigate the **short-term plasticity** behaviour featured by SWARMs. Paired-pulse facilitation (PPF) refers to a short-term form of homosynaptic facilitation in which the postsynaptic response to the second action potential is much larger relative to the first due to the accumulation of residual Ca^{2+} in the presynaptic terminal from the two pulses. The degree of facilitation is greatest when the Ca^{2+} ions are not allowed to return to the baseline concentration prior to the second stimulus, that is, when the pulse interval is kept shortest [1968 *J. Physiol.* 1968, 195, 481.]. Analogous to this, action potentials (+1.5V, pw=20ms) separated by minute pulse intervals (<50ms), triggers higher excitatory post synaptic currents (EPSCs) in the second presynaptic spike, resulting in paired pulse facilitation (PPF) indices well above 100%. The SWARMs exhibit the highest PPF index ~181% for a pulse interval of 10ms. Increasing intervals results in an exponential reduction of the facilitation indices in accordance with the Ca^{2+} residual hypothesis (Figure R3). The exponential decrease of the facilitation indices indicates the temporal dynamics of the ion relaxation mechanism. Similarly, application of presynaptic pulses of opposite polarity results in a decrease in short-term conductance or paired pulse depression (PPD). The decay of the depression curves is again exponential, similar to PPF. The devices are then damaged with a knife cut 10 μm wide. Analysis of the PPF and PPD indices after the healing process indicate good recovery of the conductance levels and restoration of the ion accumulation-relaxation mechanism (Figure R3).

Since SWARMs are utilized to implement associative learning via non-volatile weight changes, we next focus on measurements of their **long-term plasticity** behaviour. Figure R4 below shows the representative long-term potentiation (LTP) and depression (LTD) curves before damage and after the healing process, and Figure R5 shows the LTP-LTD behaviour of the SWARMs as a function of the number of training cycles, before damage and after the healing process. The weight update trace follows a similar trend before and after healing, indicating complete functional recovery and healing of the ion gel dielectric after mechanical damage. The device-to-device variations are captured by the error plots in Figure R5. In general, the LTP and LTD weight updates do indicate higher variations after damage, but the trend of the overall the weight update traces remain consistent even after severe mechanical damage to the ion gel dielectric. From a yield perspective, 19 out of the 20 samples damaged recover functionally after 24 hours of healing time.

In summary, we report the first ever healable memristive elements fashioned as artificial nociceptors and synapses to the best of our knowledge, as also acknowledged by the reviewer. *We would also like to point to the real time demonstrations of healing behaviour and associative learning shown in Supplementary Movies-VI-3. The devices heal back after damage and are able to trigger motor responses in the robot. This further validates our claims on healable neuromorphic elements.*

Figure R4. Long-term Plasticity in SWARMs. Persistent biasing of the gate voltage (please refer to the input waveforms shown in Figure S8) results in LTP and LTD, widely considered a primary mechanism for learning and memory. Representative plots of the drain current before damage and after the healing process for (A) LTP and (B) LTD.

Figure R5. Switching Endurance of SWARMs. Controlled facilitation and depression achieved in SWARMs over 500 switching transitions by applying a series of potentiating (+1.5V) and depressing (-1.5V) presynaptic spikes. Each programming/erasing step consists of 10 spikes of pulse width 500ms. This represents the cycle-to-cycle variations during programming and erasing. The error bars capture the device-to-device variations obtained from 20 devices. The graph on the left represents data from devices before damage, while the graph on the right presents data after healing.

Reviewer #3 (Remarks to the Author):

In the present manuscript “Self-Healable Memristive Neuromorphic Elements for Decentralized Sensory Signal Processing in Robotics” the authors present a sensor system exploiting memristive devices in neuromorphic nociceptors. The use a three-terminal memristive device consisting of a In₂O₃:WO₃ channel that is controlled via an gated ionic liquid. This device is used in a volatile or non-volatile manner in a spiking neuron circuit as “SWARM” device and as “STAR” device. Based on these three components a sensor system is build and attached to a robot as nociceptor. The study is quite interesting and shows novel results. It is, however, not clear what the main achievement is. According to the title the device is novel but it is not really described in the manuscript. Only with the main manuscript it is even not clear that the same device type is used for the different components of the sensory systems. The manuscript needs to be revised in order to emphasize the novelty of the approach (including a comparison to literature). In addition, the device should be described in the main text and not only the supplement.

We thank the reviewer for the positive response and commending our contribution. We also very much appreciate the reviewer’s suggestions to improve our manuscript by making more objective statements.

“The study is quite interesting and shows novel results. It is, however, not clear what the main achievement is..... The manuscript needs to be revised in order to emphasize the novelty of the approach (including a comparison to literature).”

We thank the reviewer for the comment. We would like to reiterate the most significant findings and outcomes of our work.

Novelty at the device/material level:

1. *We report the first three-terminal artificial nociceptor.* Previous reports on artificial nociceptor were based on diffusive 2-terminal memristors [Yoon, J.H. et al. 2018. *Nature Communications*, 9(1), pp.1-9.; Kim, Y. et al. 2018. *Advanced Materials*, 30(8), p.1704320.].

2. *We report the first healable neuromorphic memristive devices- artificial nociceptors and synapses.* Although sensorized artificial synapses have been demonstrated previously [*Science*, 2018, 360(6392), pp.998-1003., *Science Advances*, 2018, 4(11), p.eaat7387.], but the ability to repeatably self-heal neuromorphic circuit elements have not been demonstrated yet. We also go beyond the healing demonstrations of signal processors only at the substrate level [*Nature Electronics*, 2019, 2(2), pp.75-82.]. This ability is hitherto not demonstrated for hardware neuromorphic circuits and is timely especially with the future of electronics and robotics going soft.

Novelty at the system architecture level:

We envision a decentralized approach to sensory information processing with intelligence shifted to the location of the sensing nodes as a novel neuromorphic approach to tackle the issues of information processing and learning in robotic skins. While synaptic learning is typically relegated to the central brain in other implementations, we show for the first time how incorporating associative learning via plasticity of synapses (SWARMS) in the robotic equivalent of peripheral nervous system enables fault-tolerance (identification of noxious signal even after nociceptor damage). In comparison to the artificial afferent nerve implementations utilizing oscillators, Mott memristors [Nature Communications, 2020, 11(1),

pp.1-9.] and organic synaptic transistors [Science, 2018, 360(6392), pp.998-1003.] with physically separate signal transduction and processing, we propose a novel decentralized scheme and demonstrate decision making at the sensor node as a viable solution to address the peripheral sensory signal processing in robotics. *This is the first report of a prototypical memristive sensory signal processing platform composed of three-terminal artificial nociceptors, synapses and neurons to the best of our knowledge. The system-level implementation although simple, is again first of its kind with memristive devices and serves a proof-of-concept illustration of the proposed concept.*

In summary, this concept can be readily extended to other sensing modalities and material platforms. Since our memtransistor can be configured to operate as both a nociceptor and synapse, this allows us to facilitate build a platform with a single core device configuration unlike conventional 2-terminal memristors that require special design of diffusive and drift configurations. While still at a prototypical stage, this work lays down a novel framework for building a memristive robotic nervous system to enable artificial intelligence (AI) in robots and prostheses. We have now made substantial changes to the manuscript to improve the readability of the novelty and significant findings of our work. Please refer to pages 3-5, 19-20 in the main text.

“In addition, the device should be described in the main text and not only the supplement.”

As per the reviewer’s suggestion, we have now added explanation of the device structure, their operating modes and a brief explanation of the working mechanism to the main text with details in the supporting information. Please refer to pages 7, 12 in the main text and pages 8, 13-15 in the supporting information. We also introduce our devices explicitly now in the introduction (Page 3).

Besides these major concerns, the following comments need to be addressed in a revised version.

1) The fonts in the figures are too small and should be enlarged.

We apologize for the small fonts we presented in the earlier version. We have now increased the font sizes in several figures (e.g. Figures 2, 3, 4, S2, S3, S4) in the revised version and also split the figures between the main text and supporting information for better clarity. E.g. we have split the earlier Figure 2 now into the main text (new Figure 2) and Figure-S4 in the supporting information in this revised version. We have also moved the optical images of the ion gel from Figure 3 to Figure S17 supporting information.

2) In Fig. S1c, it is not clear why 0.2 V is a threshold. The I-V trace looks rather gradual. Moreover, the sweep rate is missing as well.

We would like to clarify that -0.2V is the voltage at which the drain current of the FET starts to increase distinctly, as can be seen from the Figure R6 below (circled portion) and hence, a threshold voltage or V_{th} for channel formation was mentioned to be $\sim -0.2V$. As per the reviewer’s suggestion, we have now added the sweeping rate to the figure and also mentioned the same in the explanation of the figure. Please refer to the supporting information Figure-S1. Please also note that since we have defined multiple thresholds in the manuscript (e.g. threshold voltage of FET, pain threshold for nociceptive response and warning threshold for relaxation response), to avoid confusion between these terms, we now denote the V_{th} for channel formation as V_{on} in the revised version.

Figure R6. Transfer Characteristics of the IWO FET.

3) The switching mechanism of the three-terminal device is not clear. Which ions are moving? Why does the ion movement leads to a resistance change (volatile and non-volatile). Is there any evidence for the switching mechanism based on ionic movement?

We thank the reviewer for the comment. In the top ion gel-gated FET configuration we present, application of positive voltage pulses to the gate terminal drives the imidazolium cations in the ion gel dielectric toward the dielectric–semiconductor interface, which in turn causes accumulation of electrons in IWO layer to form a conductive channel. Upon removal of the voltage pulse, the cations gradually drift back to their equilibrium positions, which reduces channel conductivity and the drain current gradually decreases back to the resting current, resulting in the large anticlockwise hysteresis window seen in Figure-S1C. The accumulation of electrons the IWO layer during the forward scan results in lower voltage requirements for the subsequent channel formation in the reverse scan, accounting for the anticlockwise hysteresis. This ion migration-relaxation dynamics at the semiconductor– dielectric interface defines the volatile short-term memory/plasticity behaviour in our devices and is harnessed to present the temporal dynamics of artificial nociceptors or STARs.

On persistent application of voltage pulses with higher amplitude, this electrolyte-gating approach creates additional oxygen vacancies, modulating the local electronic structure of the channel, and resulting in a non-volatile memory. Application of high electric fields have been shown to trigger both electrical and structural transitions in a wide variety of materials, including insulator-metal/ Mott transitions in SrTiO₃ [Ueno, K. et al. 2011 *Nat. Nanotechnol.* 6 (7), 408.; Gallagher, P. et al. 2015 *Nat. Commun.* 6, 6437.] and structural changes and phase transitions in VO₂ [Jeong, J. et al. 2015. *Proc. of the Nat. Acad. of Sci.*, 112(4), pp.1013-1018.], MoTe₂ [Wang, Y. et al. 2017. *Nature*, 550 (7677), 487.] and SrCoO_{2.5} [Lu, N. et al. 2017 *Nature*, 546 (7656), 124.]. Very recently, we have investigated such effects on amorphous oxides and have observed the electrolyte to act as a permeable membrane for oxygen extraction/intercalation from/into the semiconducting channel [Kulkarni, M.R. et al. 2019. *Small*, 15(27), p.1901457.].

To systematically examine field-driven vacancy creation, we present the transfer curves of the FETs in the programmed and erased states (Figure R7A). To simplify the analysis, only the forward sweeps of the transfer characteristics are taken into consideration. Constant positive biasing of the ionic top-gate at +1.5V for 10 pulses results in a negative shift of the threshold

voltage (V_{th}) and higher off (I_{off}) and on-currents (I_{on}), indicating generation of excess carriers for charge transport. Biasing with higher fields or longer durations results in a further programmed shift of V_{th} , I_{on} and I_{off} to more conductive low resistance states (LRSs), confirming this observation. Pulses of opposite polarity -1.5V (erasing) shifts back the transfer curves to the original high resistance state (HRS).

Figure R7. Switching Mechanism of SWARMS. (A) Transfer curves of the FETs in the programmed and erased states. (B) XPS analysis of the O 1s peak depicting an increase in the oxygen vacancy concentration in LRS when compared to HRS. [Kulkarni, M.R. et al. 2019. *Small*, 15(27), p.1901457.] (C) Schematic of the extraction and intercalation of oxygen from and into the semiconducting channel.

We had performed X-ray photoelectron spectroscopy (XPS) measurements on identical samples and analysed for the metal-oxygen bonding (Figure R7B) before and after application of voltage biasing. The O 1s peak was analysed to estimate the modulation of oxygen vacancies since the conduction pathways in these oxides are predominantly dictated by vacant spatially dispersed ns orbitals. The O 1s peak was deconvoluted to three individual peaks located around 530.1, 531.1, and 532.2eV [John, R. A.; 2016 *Chem. Mater.*, 28 (22), 8305–8313.]. The peak at the lowest binding energy (~530.1eV) was assigned to the oxygen atoms in the fully oxidized indium environment (lattice oxygen; M–O–M). The mid-peak at ~531.1eV was assigned to oxygen ions in the oxygen-deficient region (indicative of oxygen vacancy concentration). And the peak at high binding energy (~532.2eV) was assigned to the presence of loosely bound oxygens (adsorbed oxygen) associated with the presence of hydroxyl groups on the surface [Socratous, J. et al. 2015 *Adv. Funct. Mater.*, 25 (12), 1873–1885.]. Upon biasing the devices at +1V for 1200s, the percentage area of M-O-M peak, oxygen vacancy peak and adsorbed oxygen peak changed from 74.2% to 41.4%, from 21.2% to 46.3% and from 4.7% to 12.3% respectively. The increase in the percentage of the oxygen-deficient region indicated an increase in the concentration of oxygen vacancies for thin films upon biasing. Films biased at lower dosages (1V, 15s) did not show any difference in the M-O bonding in the XPS analyses

(data not shown). This aligns with the volatile switching we observe in STARs. Please note that the bias voltage applied for the XPS measurements on SWARMs are much stronger than that used to map the LTP-LTD weight updates. This is intentionally done so as to ensure excellent non-volatile memory retention between the time of biasing and XPS measurements, in order to pick up differences in the oxygen vacancy concentration in the HRS and LRS.

The modulation of oxygen vacancies is evident in terms of the shift in threshold voltage and increase in on-state current in the transfer characteristics curve of ionic liquid gated thin film transistor. Based on these observations we propose that prolonged application of this electric field facilitates extraction of oxygen (creation of oxygen vacancies) from the semiconducting lattice to generate additional carriers for charge transport, resulting in a permanent shift of V_{th} and an increase in I_{on} . Higher electric fields accelerate this extrusion process, shifting the V_{th} , I_{on} and I_{off} by larger amounts. Application of electric field in the opposite direction (negative biasing) intercalates oxygen back into the semiconducting channel and shifts the electrical parameters back towards the original state (Figure R7C). The weight update traces of LTP and LTD shown in Figure R8 below summarizes the switching endurance of this process.

We have now added this discussion on the working mechanism to the main text with details in the supporting information. Please refer to pages 7, 12 in the main text and pages 8, 13-15 in the supporting information.

4) The authors should add a device characterization for the non-volatile switching behavior in the supplement. What is about statistics, device-to-device variability, cycle-to-cycle variability, and endurance data?

As per the reviewer's suggestion, we have now performed additional experiments to validate the non-volatile switching behaviour of our SWARMs. Figure R7-A (please refer to the previous answer) shows the representative transfer characteristics of our FETs in the programmed and erased states. The programmed states have an average retention of ~1800 secs. Figure R8 below shows the long-term potentiation and depression behaviour of our SWARMs as a function of the number of training cycles. This represents the endurance of the switching characteristics. As evident from the figures, the weight update trace follows a programmable linear trend with low variability from cycle to cycle. The device-to-device variability is indicated within Figure R8 as error plots [data from experimental measurement of 20 devices]. We have now added this data to the supporting information. Please refer to Figure S20 in supporting information.

Figure R8. Switching Endurance of SWARMs. Controlled facilitation and depression achieved in our devices over 500 switching transitions by applying a series of potentiating (+1.5V) and depressing (-1.5V) presynaptic spikes. Each programming/erasing step consists of 10 spikes of pulse width 500ms. This represents the cycle-to-cycle variations during programming and erasing. The error bars capture the device-to-device variations obtained from 20 devices.

5) Please provide more statistics for the trends shown in Fig. S2. For 1.1 V and 1.2 V only 1 pulse seems to be shown. What is the delay between the pulses?

In Figure S2, we show a train of pulses and not a single pulse. The pulse trains adopted throughout the manuscript have a pulse width of 16ms and interval of 5ms unless explicitly mentioned otherwise. As per the reviewer's comment, we have now collected additional data on the statistics of the volatile switching of the devices. Figure R9 shows the switching endurance of 20 devices measured across 100 cycles each. Each cycle consists of 46 voltage pulses applied at the gate terminal at a constant $V_{ds}=0.1V$. The cycle-to-cycle variations are captured by the error plots, while the graph itself shows the variation between devices. As can be seen, since the switching is volatile, the switching endurance is very good and remains stable for the duration tested. We have now added this data to the supporting information. Please refer to Figure S2C in supporting information.

Figure R9. Switching Endurance of STARs. Switching endurance of 20 devices measured across 100 cycles each. Each cycle consists of 46 voltage pulses applied at the gate terminal at a constant $V_{ds}=0.1V$. The cycle-to-cycle variations are captured by the error plots, while the graph itself shows the variation between devices. As can be seen, since the switching is volatile, the switching endurance is very good and remains stable for the duration tested.

6) In Fig. S2, the authors show first data in the mA range. The relaxation behavior in Fig. S3 shows data in the nA range. How can the system adapt to 6 orders of magnitude different current response? Why is this behavior called relaxation behavior? The current seems to drop by 6 order of magnitude immediately.

We thank the reviewer for the comment. We would like to clarify that the data shown in Figure S2 and S3 are measured from the same device. The difference is that while S3 shows the current response of the memtransistor to only 2 pulses, S2 shows the current response to 46 pulses. The current accumulates over 6 orders of magnitude due to the accumulation of more and more carriers in the semiconductor in response to the enhanced electrical double layer ion accumulation at the semiconductor-dielectric interface. This behaviour remains consistent across all our devices measured across 100s of cycles as evident from Figure R9. Figure R10 shows the distribution of the ON-OFF ratio across 20 FETs, extracted from their DC I-V transfer characteristics. All devices depict good FET operation in the same voltage window. We have now added this data to the supporting information. Please refer to Figure S1E in supporting information.

Figure R10. Statistics of the FET performance. Distribution of the ON-OFF ratio of 20 FETs measured in the same V_{gs} widow (-1.5V to +1.5V, sweeping rate=0.05V/s) at a constant $V_{ds}=0.1V$.

Relaxation in nociceptors is defined as a phenomenon where innocuous triggers immediately following warning or noxious stimuli could trigger significant/nociceptive responses from devices. This is done to cater to the often repeated nature of such stimuli; i.e., innocuous triggers immediately following noxious stimuli are also considered noxious for a particular interval to overprotect the injured site. This behaviour is called relaxation since it is similar to the flux dynamics of Ca^{2+} ions in biological synapse [Abarbanel, H.D. et al. 2003. *Biological cybernetics*, 89(3), pp.214-226.]. This same definition is adopted in literature [Yoon, J.H. et al. 2018. *Nature Communications*, 9(1), pp.1-9.; Kim, Y. et al. 2018. *Advanced Materials*, 30(8), p.1704320.]. As per the reviewer’s comment, we have explained this more clearly in this revision.

We would like to first clarify that because the ion migration-relaxation kinetics in the ionic dielectric of our transistors remain valid throughout 6 orders of magnitude of current of the FET operation, it is possible for us to demonstrate such short-term memory across several magnitudes of conductance. This allows us to set a very flexible threshold for pain perception, i.e. spanning nA to mA in comparison to conventional 2-terminal memristors. In the earlier version, we had demonstrated relaxation in the nA using paired pulse measurements. To demonstrate our device is capable of portraying this behaviour in the mA scale, we have carried out additional experiments that captures the effects of both sensitization and relaxation as shown in Figure R11. Here,

- (i) the normal state responses are initially measured in STARs by application of a pulse train at $V_{gs}=1V$.
- (ii) the device is allowed to relax for 50ms.
- (iii) the relaxation state responses are measured by the application of a pulse train at $V_{gs}=0.75V$. 0.75V is chosen here because it is an innocuous stimulus, in line with our earlier definition. Because the priori stimuli were not of noxious nature ($V_{gs}=1V$), the short-term memory/relaxation effect is negligible and the subsequent current response measured at $V_{gs}=0.75V$ does not cross the pain threshold $I_{nox}=3.3mA$.

(iv) A Noxious stimulus or Injury pulse ($V_{gs}=2.5V$) is next applied on the gate terminal of the STAR (not shown in the Figure).

(v) the sensitized state responses are next measured by application of a pulse train at $V_{gs}=1V$. The pain threshold $I_{nox}=3.3mA$ is crossed as expected.

(vi) the device is allowed to relax for 50ms.

(vii) the new relaxation state responses are measured by the application of a pulse train at $V_{gs}=0.75V$. Now, since the device has experienced a noxious stimuli as its antecedent ($V_{gs}=2.5V$ -stage iv), the relaxation effect becomes very prominent and causes the current to cross the pain threshold $I_{nox}=3.3mA$ even when measured at an innocuous voltage stimulus of $V_{gs}=0.75V$.

This represents how the relaxation effects heavily depend on the nature of the priori stimuli and its interval. We hope the reviewer is now satisfied with this explanation. We have added this new data to the supporting information. Please refer to the new Figure-S5 in supporting information.

Figure R11. Relaxation behaviour of our STARs. On arrival of a noxious stimulus, the STARs enter the “relaxation” phase during which it recalibrates/increases its sensitivity to cater to the often repeated nature of such stimuli; i.e., innocuous triggers immediately following noxious stimuli are also considered noxious for a particular interval to overprotect the injured site.

7) The experiment of Fig. S3 is not clear. The authors claim to use a voltage $V < V_{th}$. Here, $+0.5V$ are used but before a threshold voltage of $-0.2V$ was mentioned.

The value of $+0.5V$ is not defined as a threshold but a very weak stimulus-

“On the other hand, on arrival of extremely weak successive stimuli ($0.5V$), the devices remain unperturbed even in their relaxation phase (Figure-S3D), reiterating the volatile threshold switching property of our devices.”

The threshold switching property here only refers to the volatile nature of switching characteristics of our devices, and not a threshold voltage. Please refer to the above answer for

explanation of Figure S3. We have now amended the supporting information accordingly to clearly indicate this.

We would like to clarify that in the earlier version of the manuscript, we had defined 3 thresholds- (i) a threshold voltage for the FET's channel formation, (ii) a threshold current level specifically for the relaxation behaviour, and (iii) a pain threshold current level for sensitized nociceptive response. We understand the confusion that arises here. Thus, we have now explicitly redefined (i) as the V_{on} for the FET's channel formation (-0.2V) , (ii) as $I_{relaxation}$ for the relaxation behaviour (21nA) and (iii) I_{nox} as the pain threshold current level for sensitized nociceptive response (3.3mA). We hope it is clear now and apologize for not having mentioned this in the earlier version of the manuscript.

8) The proposed device is a 3-terminal device; a kind of synaptic transistor. The introduction is a little bit confusing as memristive devices are commonly liked to passive 2-terminal device. Thus, the authors should clearly state from the beginning that a 3-terminal device is used.

We thank the reviewer for the comment. We understand the reviewer's concern on the confusion between the general interpretation of a memristive device as a pure 2-terminal device, whereas 3-terminal configurations are often referred to as synaptic transistors. To delineate this, we have now explicitly introduced the 3-terminal device structure and modes of device operation in the introduction of the manuscript and have also reiterated this clearly at several places in the manuscript. We have also mentioned the 3-terminal configuration and its advantages over a 2-terminal configuration in the discussion section. Please refer to pages 2-4 in the main text. Most importantly, we have now changed the title of the manuscript to "**Self-Healable Neuromorphic Memtransistor Elements for Decentralized Sensory Signal Processing in Robotics**"

9) The authors mention a drift mode and a diffusion mode of the device. Both modes should be introduced before. Drift and diffusion mode are not common knowledge.

We thank the reviewer for the comment. The volatile threshold nature of our STARS draw similarity with 2-terminal diffusive memristors in their electrical switching characteristics. Similarly, the non-volatile conductance changes of our SWARMs draw comparison to 2-terminal drift memristors [Wang, Z. et al. 2017. *Nature Materials*, 16(1), pp.101-108.]. We draw this analogy strictly speaking from the perspective of their switching characteristics. To clearly define this, we have now introduced the modes of device operation explicitly in the introduction of the manuscript and also reiterated this clearly at several places in the manuscript. Some examples of the amendments made include:

"In the gated-threshold a.k.a. diffusive mode, migration and relaxation of ions in the ionic dielectric temporarily strengthens and weakens the charge carrier accumulation in the semiconducting channel, resulting in a volatile hysteresis, as detailed in SI-Note-2." (Page 7 Main Text)

"We configure thin film memtransistors to operate as gated-memristive switches a.k.a drift mode, to functionally emulate the signal processing of a biological synapse. On persistent application of positive voltage pulses with higher amplitude, additional oxygen vacancies are created in the ultra-thin IWO semiconducting channel, modulating its local electronic structure, and resulting in a non-volatile memory." (Page 12 Main Text)

"Configured as gated-threshold switches, our STARS functionally emulate the signal processing of a biological nociceptor.....This ion migration-relaxation dynamics at the

semiconductor– dielectric interface defines the volatile short-term memory/plasticity behaviour in our devices and is harnessed to present the temporal dynamics of artificial nociceptors or STARs. This is functionally analogous to the working of conventional 2-terminal diffusive memristors and hence, we refer to this as the diffusive mode of STARs/ gated-threshold switches.” (Page 8 Supporting Information Note-2)

“Configured as gated-memristive switches, SWARMs functionally emulate the signal processing of a biological synapse.....This stoichiometric modulation of the channel defines the non-volatile long-term memory/plasticity behaviour in our devices and is harnessed to present the temporal dynamics of artificial synapses or SWARMs. This is functionally analogous to the working of conventional 2-terminal drift memristors and hence, we refer to this as the drift mode of SWARMs/ gated-memristive switches.” (Page 13 Supporting Information Note-4)

10) It is not clear how the voltage is applied to the three-terminal device in the experiments in Fig. S5.

In Figure S5 (S8 in this revised version), the initial channel conductance (G_{ds}) is read by a reading spike ($V_{read}=+0.1V$, 10ms). In our memtransistor configuration, this reading voltage is applied across the drain and source terminals or in other words $V_{read}=V_{ds}$. Next, spike patterns (V_{write}) corresponding to Figure-S5 (S8 in this revised version) are then applied between the gate (V_g) and source (V_s) terminals and the change in conductance/weight (ΔG_{ds}) is recorded as a function of the pulse interval between pre- and postsynaptic spikes. The timing difference create effect writing voltages ($f(V_{pre}-V_{post}), t$) across the device, which on crossing the threshold voltage, creates long-term weight changes in the channel. The resultant conductance change is finally read again with the V_{read} pulse. We have detailed this in SI-Note-4.

11) The device stack should be explained in more detail. It is not clear for example how the gate contact to the ionic liquid is realized.

The device stack comprises of:

- Ag as the side gate electrode. In some cases contacts to the ion gel were made directly using the probe station’s tip. No difference was observed in the device characteristics in both cases. We apologize for not having mentioned this explicitly in the earlier version.
- ion gel composed of an ionic liquid 1-ethyl-3-methylimidazolium bis(trifluoromethylsulfonyl) imide, [EMI][TFSI] housed inside a poly (vinylidene fluoride- co -hexafluoropropylene) P(VDF-HFP) matrix as the top gate dielectric.
- indium-tungsten oxide (IWO) serving as the semiconducting channel.
- ITO source and drain contacts.

Figure R12. Device schematic with side gate

Detailed explanation of the experimental details- ion gel formation, device fabrication and characterization are provided in the experimental section in the main text. Please refer to page

21 in the main text. A schematic of the device stack is also shown in Figure-S1A of the supporting information.

12) In Fig. S6 R_{mem} is shown as a two-terminal device, but is supposed to be a three-terminal one. Is this a different device as the proposed one?

We apologize for the misrepresentation of R_{mem} in the earlier version of the manuscript as a 2-terminal device—it was done to show that with a fixed gate voltage V_G , the device behaves like a resistor. We would like to clarify that R_{mem} indicated in the neuron circuit (SI-Note-5 Figure S9 in the revised version) is the same 3-terminal FET we utilize as SWARM throughout the manuscript. We have now corrected the symbol to a 3-terminal FET in the revised circuit schematic.

Reviewers' Comments:

Reviewer #2:

Remarks to the Author:

The authors have collected extensive amount of new data since the first round of review, mainly on the electrical behavior of the self-healing devices, which strengthens the device aspect of this work. As suggested in the first round of review, the work on the self-healing synapses is interesting and should be the focus of this manuscript. The new studies on the self-healing mechanism synaptic behavior should be moved to the main text, making a coherent research paper on synapses that would likely generate interest in the community. The integration of the three parts (nociceptor, neuron and synapses) is actually going to distract the readers, so are the neuron and nociceptor parts, as the novelty is not as prominent as the self-healing features from the synapse. I would suggest the authors move those data to the SI or publish later when the system level of work is mature.

Reviewer #3:

Remarks to the Author:

The authors addressed all my comments in detail. I recommend accepting the paper.

REVIEWER COMMENTS

We thank both the reviewers for the positive comments on our work. In the following point by point response, the reviewer comments are in black and our responses are in blue. All changes to the main text and supporting information are highlighted in yellow within the respective documents.

Reviewer #2 (Remarks to the Author):

The authors have collected extensive amount of new data since the first round of review, mainly on the electrical behavior of the self-healing devices, which strengthens the device aspect of this work. As suggested in the first round of review, the work on the self-healing synapses is interesting and should be the focus of this manuscript. The new studies on the self-healing mechanism synaptic behavior should be moved to the main text, making a coherent research paper on synapses that would likely generate interest in the community. The integration of the three parts (nociceptor, neuron and synapses) is actually going to distract the readers, so are the neuron and nociceptor parts, as the novelty is not as prominent as the self-healing features from the synapse. I would suggest the authors move those data to the SI or publish later when the system level of work is mature.

We thank the referee for the positive comments on our revised work and acknowledging the significance of our work on self-healing memristive and neuromorphic devices. We sincerely appreciate the reviewer's constructive feedback to further improve our manuscript. As per the reviewer's suggestion, we have now moved the new studies on the self-healing synaptic behaviour to the main text and made a separate coherent figure to depict the same. We have also added explanations to the main text. Please refer to the new Figure 5 in the main text which is reproduced below for your reference.

Figure 5 Self-Healing Memristive Neuromorphic Elements-SWARMs. (A) Short-term Plasticity. A pair of presynaptic action potentials (+1.5V, pulse width = 20ms, interval = 10ms) triggers a pair of

excitatory postsynaptic currents (EPSCs) with increasing amplitude. This phenomenon known as PPF reflects the number of residual carriers during ion migration relaxation kinetics in the ionic-gated mode. Reversal of polarity of the presynaptic action potentials (-1.5V) result in PPD with the indices dependent on pulse width and interval of the presynaptic action potentials, similar to facilitation. PPF/D indices, defined as $[PPF/D = \left(\frac{A_2}{A_1}\right) * 100\%]$ is plotted as a function of inter-spike interval to demonstrate the decay process. **(B) Long-term Plasticity.** Electrical characterizations of STDP recorded on SWARMs at various stages of the damage and healing process. **(C)** Controlled long-term facilitation and depression achieved in SWARMs over 500 switching transitions by applying a series of potentiating (+1.5V) and depressing (-1.5V) presynaptic spikes. Each programming/erasing step consists of 10 spikes of pulse width 500ms. The figure represents the cycle-to-cycle variations during programming and erasing. The error bars capture the device-to-device variations obtained from 20 devices. The LTP and LTD weight updates do indicate higher variations after damage, but the trend of the overall the weight update traces remain consistent even after severe mechanical damage to the ion gel dielectric.

Figure 5A depicts the healing behaviour of short-term plasticity of SWARMs. Paired-pulse facilitation (PPF) and depression (PPD)-features that play vital roles in signal filtering are now shown in the main text. Analysis of the PPF and PPD indices after the healing process indicate good recovery of the conductance levels and restoration of the ion accumulation-relaxation mechanism (Figure-5A, SI-Note-6 Figure-S18). Since SWARMs are utilized to implement associative learning via non-volatile weight changes, we next focus on measurements of their long-term plasticity behaviour. Figures-5B-C depicts the STDP and LTP-LTD behaviour of our SWARMs as a function of the number of training cycles, before damage and after the healing process respectively. As evident from the figures, the weight update trace follows a similar trend before and after healing, indicating complete functional recovery and healing of the ion gel dielectric after mechanical damage.

The reviewer also commented on removing the integration (last figure) and related to discussion from the main text. However we believe that the implementations of nociceptors, synapses and neurons and their robotic integration sets the context in the manuscript. Based on the reviewer's suggestion, we have indeed reduced the number of demonstration sub-figures in Figure 6 (main text)—but we feel it is important to retain at least the part of it due to the following reasons. In addition to the report of the first three-terminal artificial nociceptor, we propose a novel decentralized scheme and demonstrate decision making at the sensor node as a viable solution to address the peripheral sensory signal processing in robotics. Prior demonstrations of artificial afferent nerves [e.g. Tan H et al. Nature Communications. 2020 Mar 13;11(1):1-9.; Liao X et al. Nature Communications. 2020 Jan 14;11(1):1-9.; Zhang X et al. Nature Communications. 2020 Jan 2;11(1):1-9.] have proposed physically separate signal transduction and processing. In contrast, we propose a novel decentralized scheme based on the unique sliding threshold behaviour of nociceptors and associative learning in synapses to shift intelligence to the location of the sensor nodes for decentralized decision making. We believe this would be interesting to the interdisciplinary readership of Nature Communications.

Hence, we propose to preserve these portions in the main text and hope the reviewer agrees to the new manuscript organization.

Reviewer #3 (Remarks to the Author):

The authors addressed all my comments in detail. I recommend accepting the paper.

We thank the reviewer for the positive feedback on our revised manuscript.

Reviewers' Comments:

Reviewer #2:

Remarks to the Author:

The authors have successfully addressed the concerns from the reviewer.

REVIEWERS' COMMENTS:

Reviewer #2 (Remarks to the Author):

The authors have successfully addressed the concerns from the reviewer.

We thank the reviewer for the positive feedback on our revised manuscript.